# Safe Exploration in Reinforcement Learning: A Generalized Formulation and Algorithms

**Akifumi Wachi**
LINE Corporation
akifumi.wachi@linecorp.com

**Wataru Hashimoto**
Osaka University
hashimoto@is.eei.eng.osaka-u.ac.jp

**Xun Shen**
Osaka University
shenxun@eei.eng.osaka-u.ac.jp

**Kazumune Hashimoto**
Osaka University
hashimoto@eei.eng.osaka-u.ac.jp

## Abstract

Safe exploration is essential for the practical use of reinforcement learning (RL) in many real-world scenarios. In this paper, we present a generalized safe exploration (GSE) problem as a unified formulation of common safe exploration problems. We then propose a solution of the GSE problem in the form of a meta-algorithm for safe exploration, MASE, which combines an unconstrained RL algorithm with an uncertainty quantifier to guarantee safety in the current episode while properly penalizing unsafe explorations *before actual safety violation* to discourage them in future episodes. The advantage of MASE is that we can optimize a policy while guaranteeing with a high probability that no safety constraint will be violated under proper assumptions. Specifically, we present two variants of MASE with different constructions of the uncertainty quantifier: one based on generalized linear models with theoretical guarantees of safety and near-optimality, and another that combines a Gaussian process to ensure safety with a deep RL algorithm to maximize the reward. Finally, we demonstrate that our proposed algorithm achieves better performance than state-of-the-art algorithms on grid-world and Safety Gym benchmarks without violating any safety constraints, even during training.

## 1 Introduction

Safe reinforcement learning (RL) is a promising paradigm that enables policy optimizations for safety-critical decision-making problems (e.g., autonomous driving, healthcare, and robotics), where it is necessary to incorporate safety requirements to prevent RL policies from posing risks to humans or objects [14]. As a result, safe exploration has received significant attention in recent years as a crucial issue for ensuring the safety of RL during both the learning and execution phases [6].

Safe exploration in RL has typically been addressed by formulating a constrained RL problem in which the policy optimization is subject to safety constraints [9, 18]. While there have been many attempts under different types of constraint representations (e.g., expected cumulative cost [3], CVaR [29]), satisfying constraints almost surely or with high probability received less attention to date. Imagine safety-critical applications such as planetary exploration where even a single constraint violation may result in catastrophic failure. NASA's engineers hope Mars rovers to ensure safety at least with high probability [8]; thus, constraint satisfaction "on average" does not fit their purpose.

While several algorithms have addressed this problem with this stricter notion of safety, there are several formulations in terms of how the constraints are represented, including cumulative [32], state [39], and instantaneous constraints [41], which respectively correspond to Problems 1, 2, and 3

37th Conference on Neural Information Processing Systems (NeurIPS 2023).

as we will discuss shortly in Section 2. Unfortunately, there has been limited discussion on the relationships between these approaches, making it challenging for researchers to acquire a systematic understanding of the field as a whole. If a generalized problem were to be formulated, then the research community could pool their efforts to develop suitable algorithms.

A closer examination of existing algorithms that span the entire theory-to-practice spectrum reveals several areas for improvement. Practical algorithms using deep RL (e.g., [32],[39],[40]) may provide satisfactory performance after convergence, but do not usually guarantee safety during training. In contrast, theoretical studies (e.g., [4], [41]) that guarantee safety with high probability during training often have limitations, such as relying on strong assumptions (e.g., known state transition) or experiencing decreased performance in complex environments. In summary, many algorithms have been proposed in various safe RL formulations, but the creation of a safe exploration algorithm that is both practically useful and supported by theoretical foundations remains an open problem.

**Contributions.** We first present a generalized safe exploration (GSE) problem and prove its generality compared with existing safe exploration problems. By taking advantage of the tractable form of the safety constraint in the GSE problem, we establish a meta-algorithm for safe exploration, MASE. This algorithm employs an uncertainty quantifier for a high-probability guarantee that the safety constraints are not violated and penalizes the agent *before* safety violation, under the assumption that the agent has access to an "emergency stop" authority. Our MASE is both practically useful and theoretically well-founded, which allows us to optimize a policy via an arbitrary RL algorithm under the high-probability safety guarantee, even during training. We then provide two specific variants of MASE with different uncertainty quantifiers. One is based on generalized linear models (GLMs), for which we theoretically provide high-probability guarantees of safety and near-optimality. The other is more practical, combining a Gaussian process (GP, [27]) to ensure safety with a deep RL algorithm to maximize the reward. Finally, we show that MASE performs better than state-of-the-art algorithms on the grid-world and Safety Gym [28] without violating any safety constraints, even during training.

## 2 Preliminaries

**Definitions.** We consider an episodic safe RL problem in a constrained Markov decision process (CMDP, [3]), $\mathcal{M} = \langle \mathcal{S}, \mathcal{A}, H, \mathcal{P}, r, g, s_1 \rangle$, where $\mathcal{S}$ is a state space, $\mathcal{A}$ is an action space, $H \in \mathbb{Z}_{>0}$ is a (fixed) length of each episode, $\mathcal{P} : \mathcal{S} \times \mathcal{A} \times \mathcal{S} \to [0, 1]$ is a state transition probability, $r : \mathcal{S} \times \mathcal{A} \to [0, 1]$ is a reward function, $g : \mathcal{S} \times \mathcal{A} \to [0, 1]$ is a safety (cost) function, and $s_1 \in \mathcal{S}$ is an initial state. At each discrete time step, with a given (fully-observable) state $s$, the agent selects an action $a$ with respect to its policy $\pi : \mathcal{S} \to \mathcal{A}$, receiving the new state $s'$, reward $r$, and safety cost $g$. Though we assume a deterministic policy, our core ideas can be extended to stochastic policy settings. Given a policy $\pi$, the value and action-value functions in a state $s$ at time $h$ are respectively defined as

$$V_{r,h}^\pi(s) := \mathbb{E}_\pi \left[ \sum_{h'=h}^{H} \gamma_r^{h'} r(s_{h'}, a_{h'}) \,\middle|\, s_h = s \right]$$

and $Q_{r,h}^\pi(s,a) := \mathbb{E}_\pi \left[ \sum_{h'=h}^{H} \gamma_r^{h'} r(s_{h'}, a_{h'}) \mid s_h = s, a_h = a \right]$, where the expectation $\mathbb{E}_\pi$ is taken over the random state-action sequence $\{(s_{h'}, a_{h'})\}_{h'=h}^{H}$ induced by the policy $\pi$. Additionally, $\gamma_r \in (0, 1]$ is a discount factor for the reward function. In the remainder of this paper, we define $V_{\max} := \frac{1-\gamma_r^H}{1-\gamma_r}$ and let $\mathcal{T}_h : (\mathcal{S} \times \mathcal{A} \to \mathbb{R}) \to (\mathcal{S} \times \mathcal{A} \to \mathbb{R})$ denote the Bellman update operator $\mathcal{T}_h(Q)(s,a) := \mathbb{E}[r(s_h, a_h) + \gamma_r V_Q(s_{h+1}) \mid s_h = s, a_h = a]$, where $V_Q(s) := \max_{a \in \mathcal{A}} Q(s,a)$.

**Three common safe RL problems.** We tackle safe RL problems where constraints must be satisfied almost surely, even during training. While such problems have garnered attention in the research community, there are several types of formulations, and their relations are yet to be fully investigated.

One of the most popular formulations for safe RL problems involves maximizing $V_r^\pi := V_{r,1}^\pi(s_1)$ under the constraint that the cumulative cost is less than a threshold, which is described as follows:

**Problem 1** (Almost surely safe RL with cumulative constraint [32])**.**

$$\max_\pi V_r^\pi \quad \text{subject to} \quad \Pr \left[ \sum_{h=1}^{H} \gamma_g^h g(s_h, a_h) \le \xi_1 \,\middle|\, \mathcal{P}, \pi \right] = 1,$$

where $\xi_1 \in \mathbb{R}_{\ge 0}$ is a constant representing a threshold, and $\gamma_g \in (0, 1]$ is a discount factor for $g$.

Observe that, the expectation is *not* taken regarding the safety constraint in Problem 1. This problem was studied in [32], which is stricter than the conventional one where the expectation is taken with respect to the cumulative safety cost function (i.e., $\mathbb{E}_\pi [\sum_{h=1}^H \gamma_g^h g(s_h, a_h)] \leq \xi_1$).

Another popular formulation involves leveraging the state constraints so that safety corresponds to avoiding visits to a set of unsafe states. This type of formulation has been widely adopted by previous studies on safe-critical robotics tasks [38–40, 45], which is written as follows:

**Problem 2** (Safe RL with state constraints)**.**

$$\max_\pi V_r^\pi \quad \text{subject to} \quad \mathbb{E}\left[\sum_{h=1}^H \gamma_g^h \, \mathbb{I}(s_h \in S_{\text{unsafe}}) \,\middle|\, \mathcal{P}, \pi\right] \leq \xi_2,$$

where $\xi_2 \in \mathbb{R}_{\geq 0}$ is a threshold, $\mathbb{I}(\cdot)$ is the indicator function, and $S_{\text{unsafe}} \subset \mathcal{S}$ is a set of unsafe states.

Finally, some existing studies formulate safe RL problems via an instantaneous constraint, attempting to ensure safety even during the learning stage while aiming for extremely safety-critical applications such as planetary exploration [42] or healthcare [36]. Such studies typically require the agent to satisfy the following instantaneous safety constraint at every time step.

**Problem 3** (Safe RL with instantaneous constraints)**.**

$$\max_\pi V_r^\pi \quad \text{subject to} \quad \Pr\Big[g(s_h, a_h) \leq \xi_3 \mid \mathcal{P}, \pi\Big] = 1, \quad \forall h \in [\, 1, H \,],$$

where $\xi_3 \in \mathbb{R}_{\geq 0}$ is a time-invariant safety threshold.

# 3   Problem Formulation

This paper also requires an agent to optimize a policy under a safety constraint, as in the three common safe RL problems. We seek to find the optimal policy $\pi^\star : \mathcal{S} \to \mathcal{A}$ of the following problem, which will hereinafter be referred to as the "generalized" safe exploration (GSE) problem:

**Problem 4** (GSE problem)**.** Let $b_h \in \mathbb{R}$ denote a time-varying threshold.

$$\max_\pi V_r^\pi \quad \text{subject to} \quad \Pr\Big[g(s_h, a_h) \leq b_h \mid \mathcal{P}, \pi\Big] = 1, \quad \forall h \in [\, 1, H \,].$$

This constraint is instantaneous, which requires the agent to learn a policy without a single constraint violation not only after convergence but also during training. We assume that the threshold is myopically known; that is, $b_h$ is known at time $h$, but unknown before that. Crucially, at every time step $h$, since $s_h$ is a fully observable state and the agent's policy is deterministic, we will use a simplified inequality represented as $g(s_h, a_h) \leq b_h$ in the rest of this paper. This constraint is akin to that in Problem 3, with the difference that the safety threshold is time-varying.

**Importance of the GSE problem.** Though our problem may not seem relevant to Problems 1 and 2, we will shortly present and prove a theorem on the relationship between the GSE problem and the three common safe RL problems.

**Theorem 3.1.** *Problems 1, 2, and 3 can be transformed into the GSE problem (i.e., Problem 4).*

See Appendix B for the proof. In other words, the feasible policy space in the GSE problem can be identical to those in the other three problems by properly defining the safety cost function $g$ and threshold $b_h$. Crucially, Problem 1 is a special case of the GSE problem with $b_h = \eta_h$ for all $h$, where $\eta_{h+1} = \gamma_g^{-1} \cdot (\eta_h - g(s_h, a_h))$ with $\eta_0 = \xi_1$. It is particularly beneficial to convert Problems 1 and 2, which have additive constraint structures, to the GSE problem, which has an instantaneous constraint. The accurate estimation of the cumulative safety value in Problems 1 and 2 is difficult because they depend on the trajectories induced by a policy. Dealing with the instantaneous constraint in the GSE problem is easier, both theoretically and empirically. Also, especially when the environment is time-varying (e.g., there are moving obstacles), the GSE problem is more useful than Problem 3.

Typical CMDP formulations with expected cumulative (safety) cost are out of the scope of the GSE problem. In such problems, the safety notion is milder; hence, although many advanced deep RL algorithms have been actively proposed that perform well in complex environments after convergence,

their performance in terms of safety during learning is usually low, as reported by Stooke et al. [33] or Wachi et al. [43]. Risk-constrained MDPs are also important safe RL problems that are *not* covered by the GSE problem; they have been widely studied by representing risk as a constraint on some conditional value-at-risk [11] or using chance constraints [24, 26].[1]

**Difficulties and Assumptions.** Theorem 3.1 insists that the GSE problem covers a wide range of safe RL formulations and is worth solving, but the problem is intractable without assumptions. We now discuss the difficulties in solving the GSE problem, and then list the assumptions in this paper.

The biggest difficulty with the GSE problem lies in the fact that there may be no viable safe action given the current state $s_h$, safety cost $g$, and threshold $b_h$. When $b_h = 0.1$ and $g(s_h, a) = 0.5, \forall a \in \mathcal{A}$, the agent has no viable action for ensuring safety. The agent needs to guarantee safety, even during training, where little environmental information is available; hence, it is significant for the agent to avoid such situations where there is no action that guarantees safety. Another difficulty is related to the regularity of the safety cost function and the strictness of the safety constraint. In this paper, the safety cost function is unknown a priori.; thus, when the safety cost does not exhibit any regularity, the agent can neither infer the safety of decisions nor guarantee safety almost surely.

To address the first difficulty mentioned above, we use Assumptions 3.2 and 3.3.

**Assumption 3.2** (Safety margin). There exists $\zeta \in \mathbb{R}_{>0}$ such that $\Pr[\, g(s_h, a_h) \leq b_h - \zeta \mid \mathcal{P}, \pi^\star\,] = 1, \forall h \in [1, H]$.

**Assumption 3.3** (Emergency stop action). Let $\widehat{a}$ be an emergency stop action such that $\mathcal{P}(s_1 \mid s, \widehat{a}) = 1$ for all $s \in \mathcal{S}$. The agent is allowed to execute the emergency stop action and reset the environment if and only if the probability of guaranteed safety is not sufficiently high.

Assumption 3.2 is mild; this is similar to the Slater condition, which is widely adopted in the CMDP literature [13, 25]. We consider Assumption 3.3 is also natural for safety-critical applications because it is usually better to guarantee safety, even with human interventions, if the agent requires help in emergency cases. In some applications (e.g., the agent is in a hazardous environment), however, emergency stop actions should often be avoided because of the expensive cost of human intervention. In such cases, the agent needs to learn a reset policy allowing them to return to the initial state as in Eysenbach et al. [15], rather than asking for human help, which we will leave to future work.

As for the second difficulty, we assume that the safety cost function belongs to a class where uncertainty can be estimated and guarantee the satisfaction of the safety constraint with high probability. We present an assumption regarding an important notion called an uncertainty quantifier:

**Assumption 3.4** ($\delta$-uncertainty quantifier). Let $\mu : \mathcal{S} \times \mathcal{A} \to \mathbb{R}$ denote the estimated mean function of safety. There exists a $\delta$-uncertainty quantifier $\Gamma : \mathcal{S} \times \mathcal{A} \to \mathbb{R}$ such that $\mid g(s, a) - \mu(s, a) \mid \leq \Gamma(s, a)$ for all $(s, a) \in \mathcal{S} \times \mathcal{A}$, with a probability of at least $1 - \delta$.

## 4 Method

We propose MASE for the GSE problem, which combines an unconstrained RL algorithm with additional mechanisms for addressing the safety constraints. The pseudo-code is provided in Algorithm 1, and a conceptual illustration can be seen in Figure 1.

The most notable feature of MASE is that safety is guaranteed via the $\delta$-uncertainty quantifier and the emergency stop action (lines $3 - 9$). The $\delta$-uncertainty quantifier is particularly useful because we can guarantee that the confidence bound contains the true safety cost function, that is, $g(s, a) \in [\, \mu(s, a) \pm \Gamma(s, a)\,]$ for all $s \in \mathcal{S}$ and $a \in \mathcal{A}$. This means that, if the agent chooses actions such that $\mu(s_h, a_h) + \Gamma(s_h, a_h) \leq b_h$, then $g(s_h, a_h) \leq b_h$ holds with a probability of at least $1 - \delta$. Regarding the first difficulty mentioned in Section 3, it is crucial that there is at least one safe action. Thus, at every time step $h$, the agent computes a set of actions that are considered to satisfy the safety constraints with a probability at least $1 - \delta$ given the state $s_h$ and threshold $b_h$. This is represented as

$$\mathcal{A}_h^+ := \{\, a \in \mathcal{A} \mid \min\{\, 1, \mu(s_h, a) + \Gamma(s_h, a)\,\} \leq b_h \,\}.$$

Whenever the agent identifies that at least one action will guarantee safety, the agent is required to choose an action within $\mathcal{A}_h^+$ (line 3). The emergency stop action is executed if and only if there is no

---

[1]The solution in the GSE problem is guaranteed to be a conservative approximation of that in safe RL problems with chance constraints. For more details, see Appendix C.

---
**Algorithm 1** Meta-Algorithm for Safe Exploration (MASE)
---
1: **for** episode $t = 1, 2, \ldots, T$ **do**
2:      **for** time $h = 1, 2, \ldots, H$ **do**
3:          Take "safe" action $a_h = \pi(s_h)$ within $\mathcal{A}_h^+$             $\triangleright$ Execute only safe actions
4:          Receive reward $r(s_h, a_h)$, safety cost $g(s_h, a_h)$, and next state $s_{h+1}$
5:          Update safety threshold $b_{h+1}$
6:          **if** $\mathcal{A}_{h+1}^+ = \emptyset$ **then**
7:             Compute $\widehat{r}(s_h, a_h) = -\frac{c}{\min_{a \in \mathcal{A}} \Gamma(s_{h+1}, a)}$     $\triangleright$ Penalty for the emergency stop action
8:             Append $(s_h, a_h, \widehat{r}(s_h, a_h), s_{h+1})$ to $\mathcal{D}$
9:             **break** (i.e., take action $\widehat{a}$)           $\triangleright$ Execute the emergency stop action
10:          **else**
11:             Append $(s_h, a_h, r(s_h, a_h), s_{h+1})$ to $\mathcal{D}$
12:      Optimize a policy $\pi$ based on $\mathcal{D}$ via an (unconstrained) RL algorithm
13:      Update the uncertainty quantifier $\Gamma$ and rewrite $\mathcal{D}$
---

viable action satisfying the safety constraint (i.e., $\mathcal{A}_{h+1}^+ \neq \emptyset$); that is, the agent is allowed to execute $\widehat{a}$ and start a new episode from an initial safe state (lines $6-9$). The safety cost is upper-bounded by 1 because $g \in [0, 1]$ by definition. Note that MASE proactively avoids unsafe actions by selecting the emergency stop action to take *beforehand*; this is in contrast to Sun et al. [37], whose method terminates the episode immediately *after* the agent has already violated a safety constraint.

When safety is guaranteed in the manner described above, the question remains as to how to obtain a policy that maximizes the expected cumulative reward. As such, we first convert the original CMDP $\mathcal{M}$ to the following unconstrained MDP

$$\widehat{\mathcal{M}} \coloneqq \langle\, \mathcal{S}, \{\mathcal{A}, \widehat{a}\}, H, \mathcal{P}, \widehat{r}, s_1 \,\rangle.$$

The changes from $\mathcal{M}$ lie in the action space and the reward function, as well as in the absence of the safety cost function. First, the action space is augmented so that the agent can execute the emergency stop action, $\widehat{a}$. The second modification concerns the reward function. When executing the emergency stop action $\widehat{a}$, the agent is penalized as its sacrifice so that the same situation will not occur in future episodes; hence, we modify the reward function as follows:

$$\widehat{r}(s_h, a_h) = \begin{cases} -\,c/\min_{a \in \mathcal{A}} \Gamma(s_{h+1}, a) & \text{if } \mathcal{A}_{h+1}^+ = \emptyset, \\ r(s_h, a_h) & \text{otherwise,} \end{cases} \tag{1}$$

where $c \in \mathbb{R}_{>0}$ is a positive scalar representing a penalty for performing the emergency stop. This penalty is assigned to the state-action pair $(s_h, a_h)$ that placed the agent into the undesirable situation at time step $h + 1$, represented as $\mathcal{A}_{h+1}^+ = \emptyset$ (see Figure 1).

To show that MASE is a reasonable safe RL algorithm, we express the following intuitions. Consider the ideal situation in which the safety cost function is accurately estimated for any state-action pairs; that is, $\Gamma(s, a) = 0$ for all $(s, a)$. In this case, all emergency stop actions are properly executed, and the safety constraint will be violated at the next time step if the agent executes other actions. It is reasonable for the agent to receive a penalty of $\widehat{r}(s, a) = -\infty$ because this state-action pair surely causes a safety violation without the emergency stop action. Unfortunately, however, the safety

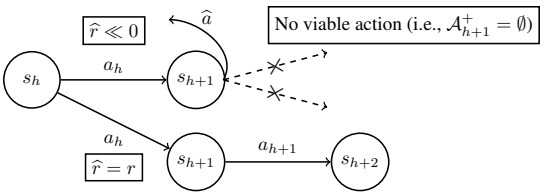

Figure 1: Conceptual illustration of MASE. At every time step $h$, the agent chooses action $a_h$ within $\mathcal{A}_h^+$. If there is no safe action at state $s_{h+1}$ satisfying the constraint, the emergency stop action $\widehat{a}$ is executed and the agent receives a large penalty for $(s_h, a_h)$.

cost is uncertain and the agent conservatively executes $\widehat{a}$ although there are still actions satisfying the safety constraint, especially in the early phase of training; hence, we increase or reduce the penalty according to the magnitude of uncertainty in (1) to avoid an excessively large penalty.

We must carefully consider the fact that the quality of information regarding the modified reward function $\widehat{r}$ is uneven in the replay buffer $\mathcal{D}$. Specifically, in the early phase of training, the $\delta$-

uncertainty quantifier is loose. Hence, the emergency stop action is likely to be executed even if viable actions remain; that is, the agent will receive unnecessary penalties. In contrast, the emergency stop actions in later phases are executed with confidence, as reflected by the tight $\delta$-uncertainty quantifier. Thus, as in line 15, we rewrite the replay buffer $\mathcal{D}$ while updating $\Gamma$ depending on the model in terms of the safety cost function (as for specific methods to update $\Gamma$, see Sections 5 and 6).

**Connections to shielding methods.** The notion of the emergency stop action is akin to shielding [2, 20] which has been actively studied in various problem settings including partially-observable environments [10] or multi-agent settings [23]. Thus, MASE can be regarded as a variant of shielding methods (especially, preemptive shielding in [2]) that is specialized for the GSE problem. On the other hand, MASE does not only block unsafe actions but also provides proper penalties for executing the emergency stop actions based on the uncertainty quantifier, which leads to rigorous theoretical guarantees presented shortly. Such theoretical advantages can be enjoyed in many safe RL problems because of the wide applicability of the GSE problem backed by Theorem 3.1.

**Advantages of MASE.** Though certain existing algorithms for Problem 3 (i.e., the closest problem to the GSE problem) theoretically guarantee safety during learning, several strong assumptions are needed, such as a known and deterministic state transition and regular safety function as in [41] and a known feature mapping function that is linear with respect to transition kernels, reward, and safety as in [5]. Such algorithms have little affinity with deep RL; thus, their actual performance in complex environments tends to be poor. In contrast, MASE is compatible with any advanced RL algorithms, which can also handle various constraint formulations while maintaining the safety guarantee.

**Validity of MASE.** We conclude this section by presenting the following two theorems to show that our MASE produces reasonable operations in solving the GSE problem.

**Theorem 4.1.** *Under Assumption 3.4, MASE guarantees safety with a probability of at least* $1 - \delta$.

**Theorem 4.2.** *Assume that the safety cost function is estimated for any state-action pairs with an accuracy of better than* $\frac{\varsigma}{2}$*; that is,* $\Gamma(s, a) \leq \frac{\varsigma}{2}$ *for all* $(s, a)$*. Set* $c \in \mathbb{R}$ *to be a sufficiently large scalar such that* $c > \frac{\varsigma V_{\max}}{2 \gamma_r^H}$*. Then, the optimal policy in* $\widehat{\mathcal{M}}$ *is identical to that in* $\mathcal{M}$.

See Appendix D for the proofs. Unfortunately, obtaining a $\delta$-uncertainty qualifier that works in general cases is highly challenging. To develop a feasible model for the uncertainty quantification, we assume that the safety cost can be modeled via a GLM in Section 5 and via a GP in Section 6.

# 5 A Provable Algorithm under Generalized Linear CMDP Assumptions

In this section, we focus on CMDPs with generalized linear structures and analyze the theoretical properties of MASE. Specifically, we provide a provable algorithm to use a class of GLMs denoted as $\mathcal{F}$ for modeling $Q_{r,h}^{\star} := Q_{r,h}^{\pi^{\star}}$ and $g$, and then provide theoretical results on safety and optimality.

## 5.1 Generalized Linear CMDPs

We extend the assumption in Wang et al. [44] from unconstrained MDPs to CMDPs settings. Our assumption is based on GLMs as with [44] that makes a strictly weaker assumption than their Linear MDP assumption [19, 46]. As preliminaries, we first list the necessary definitions and assumptions.

**Definition 5.1** (GLMs). Let $d \in \mathbb{Z}_{>0}$ be a feature dimension and let $\mathbb{B}^d := \{\mathbf{x} \in \mathbb{R}^d : \|\mathbf{x}\|_2 \leq 1\}$ be the $l_2$ ball in $\mathbb{R}^d$. For a known feature mapping function $\phi : \mathcal{S} \times \mathcal{A} \to \mathbb{B}^d$ and a known link function $f : [-1, 1] \to [-1, 1]$, the class of generalized linear model is denoted as $\mathcal{F} := \{(s, a) \to f(\langle \phi_{s,a}, \theta \rangle) : \theta \in \mathbb{B}^d\}$ where $\phi_{s,a} := \phi(s, a)$.

**Assumption 5.2** (Regular link function). The link function $f(\cdot)$ is twice differentiable and is either monotonically increasing or decreasing. Furthermore, there exist absolute constants $0 < \underline{\kappa} < \overline{\kappa} < \infty$ and $M < \infty$ such that $\underline{\kappa} < |f'(\mathbf{x})| < \overline{\kappa}$ and $|f''(\mathbf{x})| < M$ for all $|\mathbf{x}| \leq 1$.

This assumption on the regular link function is standard in previous studies (e.g., [16], [21]). Linear and logistic models are the special cases of the GLM where the link functions are defined as $f(\mathbf{x}) = \mathbf{x}$ and $f(\mathbf{x}) = 1/(1 + e^{-\mathbf{x}})$. In both cases, the link functions satisfy Assumption 5.2.

We finally make the assumption of generalized linear CMDPs (GL-CMDPs), which extends the notion of the optimistic closure for unconstrained MDP settings in Wang et al. [44].

**Assumption 5.3** (GL-CMDP). *For any $1 \le h < H$ and $u \in \mathcal{F}_{up}$, we have $\mathcal{T}_h(u) \in \mathcal{F}$ and $g \in \mathcal{F}$.*

Recall that $\mathcal{T}_h$ is the Bellman update operator. In Assumption 5.3, with a positive semi-definite matrix $A \in \mathbb{R}^{d \times d} \succ 0$ and a fixed positive constant $\alpha_{max} \in \mathbb{R}_{>0}$, we define

$$\mathcal{F}_{up} := \{(s,a) \to \min\{V_{max}, f(\langle \phi_{s,a}, \theta \rangle) + \alpha \|\phi_{s,a}\|_A\} : \theta \in \mathbb{B}^d, 0 \le \alpha \le \alpha_{max}, \|A\|_{op} \le 1\},$$

where $\|\mathbf{x}\|_A := \sqrt{\mathbf{x}^\top A \mathbf{x}}$ is the matrix Mahalanobis seminorm, and $\|A\|_{op}$ is the matrix operator norm. For simplicity, we suppose the same link functions for the Q-function and the safety cost function, but it is acceptable to use different link functions. Note that Assumption 5.3 is a more general assumption than Amani et al. [5] that assumes linear transition kernel, reward, and safety cost functions or Wachi et al. [43] that assumes a known transition and GLMs in terms of reward and safety cost functions.

## 5.2 GLM-MASE Algorithm

We introduce an algorithm GLM-MASE under Assumptions 5.2 and 5.3. Hereinafter, we explicitly denote the episode for each variable. For example, we let $s_h^{(t)}$ or $a_h^{(t)}$ denote a state or action at the time step $h$ of episode $t$. We also let $\widetilde{\phi}_h^{(t)} := \phi(s_h^{(t)}, a_h^{(t)})$ for more concise notations.

**Uncertainty quantifiers.** To actualize MASE in the generalized linear CMDP settings, we first need to consider how to obtain the $\delta$-uncertainty quantifier in terms of the safety cost function. Since we assume $g \in \mathcal{F}$, we can define the $\delta$-uncertainty quantifier based on the existing studies on GLMs, especially in the field of multi-armed bandit [22, 16]. Based on Assumptions 5.2 and 5.3, we now provide a lemma regarding the $\delta$-uncertainty quantifier on safety.

**Lemma 5.4.** *Suppose Assumptions 5.2 and 5.3 hold. Set $\delta = \frac{1}{TH}$. With a universal constant $C \in \mathbb{R}_{>0}$, let $C_g := C \overline{\kappa} \underline{\kappa}^{-1} \sqrt{1 + M + \overline{\kappa} + d^2 \ln \left( \frac{1 + \overline{\kappa} + \alpha_{max}}{\delta} \right)}$. Define*

$$\Gamma(s,a) := C_g \cdot \|\phi_{s,a}\|_{\Lambda_{h,t}^{-1}} \quad \text{with} \quad \Lambda_{h,t} := \sum_{\tau \le t} \widetilde{\phi}_h^{(\tau)} \widetilde{\phi}_h^{(\tau)} + I,$$

*where $I$ is the identity matrix. Let $\widehat{\theta}_{h,t}^g \in \mathbb{R}^d$ be the ridge estimate, which is computed by $\widehat{\theta}_{h,t}^g :=$ $\arg\min_{\|\theta\|_2 \le 1} \sum_{\tau \le t} \left( g(s_h^{(\tau)}, a_h^{(\tau)}) - f(\langle \widetilde{\phi}_h^{(\tau)}, \theta \rangle) \right)^2$. Then, the following inequality holds*

$$|g(s_h^{(t)}, a_h^{(t)}) - f(\langle \widetilde{\phi}_h^{(t)}, \widehat{\theta}_{h,t}^g \rangle)| \le \Gamma(s_h^{(t)}, a_h^{(t)})$$

*for all $(s,a) \in \mathcal{S} \times \mathcal{A}$, with a probability at least $1 - \delta$.*

For the purpose of qualifying uncertainty in GLMs, the weighted $l_2$-norm of $\phi$ (i.e., $\|\phi_{s,a}\|_{\Lambda_{h,t}^{-1}}$) plays an important role. Because we assume that the Q-function and safety cost function share the same feature, we have a similar lemma on the uncertainty quantifier regarding the Q-function as follows:

**Lemma 5.5.** *Suppose Assumptions 5.2 and 5.3 hold. Let $\widehat{\theta}_{h,t}^Q \in \mathbb{R}^d$ denote the ridge estimate; that is, $\widehat{\theta}_{h,t}^Q := \arg\min_{\|\theta\|_2 \le 1} \sum_{\tau \le t} \left( y_h^{(\tau)} - f(\langle \widetilde{\phi}_h^{(\tau)}, \theta \rangle) \right)^2$, where $y_h^{(\tau)} := r(s_h^{(\tau)}, a_h^{(\tau)}) + \max_{a' \in \mathcal{A}} \widehat{Q}_{r,h+1}^{(\tau)}(s_{h+1}^{(\tau)}, a')$ for all $\tau \le t$ with*

$$\widehat{Q}_{r,h}^{(t)}(s,a) := \min \left\{ V_{max}, f(\langle \phi_{s,a}, \widehat{\theta}_{h,t}^Q \rangle) + C_{Q/g} \Gamma(s,a) \right\}$$

*that is initialized with $\widehat{Q}_{r,h}^{(0)} = 0$ for all $h \le H$ and $\widehat{Q}_{r,H+1}^{(t)} = 0$ for all $1 \le t \le T$. Then, with a universal constant $C_{Q/g} \in \mathbb{R}_{>0}$, the following inequalities holds*

$$|Q_{r,h}^\star(s_h^{(t)}, a_h^{(t)}) - f(\langle \widetilde{\phi}_h^{(t)}, \widehat{\theta}_{h,t}^Q \rangle)| \le C_{Q/g} \cdot \Gamma(s_h^{(t)}, a_h^{(t)})$$

*for all $(s,a) \in \mathcal{S} \times \mathcal{A}$, with a probability at least $1 - \delta$.*

Note that $\Gamma(s_h^{(t)}, a_h^{(t)})$ is the $\delta$-uncertainty quantifier with respect to the safety cost function. One of the biggest advantages of the generalized linear CMDPs is that the magnitude of uncertainty for the Q-function is proportional to that for the safety cost function. Hence, by exploring the Q-function

based on the optimism in the face of the uncertainty principle [7, 35], the safety cost function is also explored simultaneously, which contributes to the efficient exploration of state-action spaces.

**Integration into MASE.** The GLM-MASE is an algorithm to integrate the $\delta$-uncertainty quantifiers inferred by the GLM into the MASE sequence. Detailed pseudo code is presented in Appendix E.

To deal with the safety constraint, GLM-MASE leverages the upper bound inferred by the GLM; that is, for all $h$ and $t$, the agent takes only actions that satisfy

$$f\left(\langle \widetilde{\phi}_h^{(t)}, \widehat{\theta}_{h,t}^g \rangle\right) + \Gamma\left(s_h^{(t)}, a_h^{(t)}\right) \leq b_h.$$

By Lemma 5.4, such state-action pairs satisfy the safety constraint, i.e. $g(s_h^{(t)}, s_h^{(t)}) \leq b_h$, for all $h$ and $t$, with a probability at least $1 - \delta$. If there is no action satisfying the safety constraint (i.e., $\mathcal{A}_h^+ = \emptyset$), the emergency stop action $\widehat{a}$ is taken, and then the agent receives a penalty defined in (1).

As for policy optimization, we follow the optimism in the face of the uncertainty principle. Specifically, the policy $\pi$ is optimized so that the upper-confidence bound of the Q-function characterized by $\widehat{r}$ is maximized; that is, for any state $s \in \mathcal{S}$, the policy is computed as follows:

$$\pi_h^{(t)}(s) = \arg\max_{a \in \mathcal{A}} \widehat{Q}_{\widehat{r},h}^{(t)}(s, a).$$

Intuitively, this equation enables us to 1) solve the exploration and exploitation dilemma by incorporating the optimistic estimates of the Q-function and 2) make the agent avoid generating trajectories to violate the safety constraint via the modified reward function.

**Theoretical results.** We now provide two theorems regarding safety and near-optimality. For both theorems, see Appendix E for the proofs.

**Theorem 5.6.** *Suppose the assumptions in Lemma 5.4 hold. Then, the GLM-MASE satisfies $g(s_h^{(t)}, a_h^{(t)}) \leq b_h$ for all $t \in [1, T]$ and $h \in [1, H]$, with a probability at least $1 - \delta$.*

**Theorem 5.7.** *Suppose the assumptions in Lemmas 5.4 and 5.5 hold. Let $C_1$ and $C_2$ be positive, universal constants. Also, with a sufficiently large $T$, let $t^\star$ denote the smallest integer satisfying $\lambda_{\min}(\Sigma)tH - C_1\sqrt{tHd} - C_2\sqrt{tH\ln\delta^{-1}} \geq 2C_g \cdot \zeta^{-1}$, where $\lambda_{\min}(\Sigma)$ is the minimum eigenvalue of the second moment matrix $\Sigma$. Then, the policy $\pi^{(t)}$ obtained by GLM-MASE at episode $t$ satisfies*

$$\sum_{t=t^\star}^{T} \left[V_r^{\pi^\star} - V_r^{\pi^{(t)}}\right] \leq \widetilde{O}\left(H\sqrt{d^3(T - t^*)}\right)$$

*with probability at least $1 - \delta$.*

Theorem 5.6 shows that the GLM-MASE guarantees safety with high probability for every time step and episode, which is a variant of Theorem 4.1 under the generalized linear CMDP assumption and corresponding $\delta$-uncertainty quantifier. Theorem 5.7 demonstrates the agent's ability to act near-optimally after a sufficiently large number of episodes. The proof is based on the following idea. After $t^\star$ episodes, the safety cost function and the Q-function are estimated with an accuracy better than $\frac{\zeta}{2}$. Then, based on Theorem 4.2, the optimal policy in $\widehat{\mathcal{M}}$ is identical to that in $\mathcal{M}$; thus, the agent achieves a near-optimal policy by leveraging the (well-estimated) optimistic Q-function.

## 6 A Practical Algorithm

Though we established an algorithm backed by theory under the generalized linear CMDP assumption in Section 5, it is often challenging to obtain proper feature mapping functions in complicated environments. Thus, in this section, we propose a more practical algorithm combining a GP-based estimator to guarantee safety with unconstrained deep RL algorithms to maximize the reward.

**Guaranteeing safety via GPs.** As shown in the previous sections, the $\delta$-uncertainty quantifier plays a critical role in MASE. To qualify the uncertainty in terms of the safety cost function $g$, we consider modeling it as a GP: $g(\boldsymbol{z}) \sim \mathcal{GP}(\mu(\boldsymbol{z}), k(\boldsymbol{z}, \boldsymbol{z}'))$, where $\boldsymbol{z} := [s, a]$, $\mu(\boldsymbol{z})$ is a mean function, and $k(\boldsymbol{z}, \boldsymbol{z}')$ is a covariance function. The posterior distribution over $g(\cdot, \cdot)$ is computed based on $n \in \mathbb{Z}_{>0}$ observations at state-action pairs $(\boldsymbol{z}_1, \boldsymbol{z}_2, \ldots, \boldsymbol{z}_n)$ with safety measurements $\boldsymbol{y}_n := \{y_1, y_2, \ldots, y_n\}$, where $y_n := g(\boldsymbol{z}_n) + N_n$ and $N_n \sim \mathcal{N}(0, \omega^2)$ is zero-mean Gaussian noise

with a standard deviation of $\omega \in \mathbb{R}_{\geq 0}$. We consider episodic RL problems, and so $n \approx tH + h$ for episode $t$ and time step $h$, although the equality does not hold because of the episode cutoffs. Using the past measurements, the posterior mean, variance, and covariance are computed analytically as $\mu_n(\boldsymbol{z}) = \boldsymbol{k}_n^\top(\boldsymbol{z})(\boldsymbol{K}_n + \omega^2 \boldsymbol{I})^{-1}\boldsymbol{y}_n$, $\sigma_n(\boldsymbol{z}) = k_n(\boldsymbol{z}, \boldsymbol{z})$, and $k_n(\boldsymbol{z}, \boldsymbol{z}') = k(\boldsymbol{z}, \boldsymbol{z}') - \boldsymbol{k}_n^\top(\boldsymbol{z})(\boldsymbol{K}_n + \omega^2 \boldsymbol{I})^{-1}\boldsymbol{k}_n(\boldsymbol{z}')$, where $\boldsymbol{k}_n(\boldsymbol{z}) = [k(\boldsymbol{z}_1, \boldsymbol{z}), \dots, k(\boldsymbol{z}_n, \boldsymbol{z})]^\top$ and $\boldsymbol{K}_n$ is the positive definite kernel matrix. We now present a theorem on the safety guarantee.

**Theorem 6.1.** *Assume* $\|g\|_k^2 \leq B$ *and* $N_n \leq \omega$ *for all* $n \geq 1$. *Set* $\beta_n^{1/2} \coloneqq B + 4\omega\sqrt{\nu_n + 1 + \ln(1/\delta)}$ *and construct the* $\delta$*-uncertainty quantifier by*

$$\Gamma(s, a) \coloneqq \beta_n \cdot \sigma_n(s, a), \quad \forall (s, a) \in \mathcal{S} \times \mathcal{A}, \tag{2}$$

*where* $\nu_n$ *is the information capacity associated with kernel* $k$. *Then, MASE based on* (2) *satisfies the safety constraint* $g(s_h^{(t)}, a_h^{(t)}) \leq b_h$ *for all* $t$ *and* $h$ *with a probability of at least* $1 - \delta$.

See Appendix F for the proofs. Theorem 6.1 guarantees that the safety constraint is satisfied by combining the GP-based $\delta$-uncertainty quantifier in (2) and the emergency stop action.

**Maximizing reward via deep RL.** The remaining task is to optimize the policy via the modified reward function $\widehat{r}$ in (1), whereby the agent is penalized for emergency stop actions. This problem is decoupled from the safety constraint and can be solved as the following unconstrained RL problem:

$$\pi \coloneqq \arg\max_\pi V_{\widehat{r}}^\pi. \tag{3}$$

There are many excellent algorithms for solving (3) such as trust region policy optimization (TRPO, [31]) and twin delayed deep deterministic policy gradient (TD3, [17]). One of the key benefits of our MASE is such compatibility with a broad range of unconstrained (deep) RL algorithms.

## 7 Experiments

We conduct two experiments. The first is on Safety Gym [28], where an agent must maximize the expected cumulative reward under a safety constraint with additive structures as in Problems 1 and 2. The safety cost function $g$ is binary (i.e., 1 for an unsafe state-action pair and 0 otherwise), and the safety threshold is set to $\xi_1 = 20$. The reason for choosing Safety Gym is that this benchmark is complex and elaborate, and has been used to evaluate a variety of excellent algorithms. The second is a grid world where a safety constraint is instantaneous as in Problem 3. Due to the page limit, we present the settings and results of the grid-world experiment in Appendix H.

To solve the Safety Gym tasks, we implement the practical algorithm presented in Section 6 as follows. First, we convert the problem into a GSE problem by defining $b_h \coloneqq \gamma_g^{-1} \cdot (\xi_1 - \sum_{h'=0}^{h-1} \gamma_g^{h'} g(s_{h'}, a_{h'}))$ and enforcing the safety constraint represented as $g(s_h, a_h) \leq b_h$ for every time step $h$ in each episode. Second, to infer the safety cost, we use deep GP to conduct training and inference, as in [30] when dealing with high-dimensional input spaces. Finally, as for the policy optimization in $\widehat{\mathcal{M}}$, we leverage the TRPO algorithm. We delegate other details to Appendix G.

**Baselines and metrics.** We use the following four algorithms as baselines. The first is TRPO, which is a safety-agnostic deep RL algorithm that purely optimizes a policy without safety consideration. The second and third are CPO [1] and TRPO-Lagrangian [28], which are well-known algorithms for solving CMDPs. The final algorithm is Sauté RL [32], which is a recent, state-of-the-art algorithm for solving safe RL problems where constraints must be satisfied almost surely. We employ the following three metrics to evaluate our MASE and the aforementioned four baselines: 1) the expected cumulative reward, 2) the expected cumulative safety, and 3) the maximum cumulative safety. We execute each algorithm with five random seeds and compute the means and confidence intervals.

**Results.** The experimental results are summarized in Figure 2. The figures show that TRPO, TRPO-Lagrangian, and CPO successfully learn the policies, but violate the safety constraints during training and even after convergence. Sauté RL is much safer than those three algorithms, but the safety constraint is not satisfied in some episodes, and the performance of the policy significantly deteriorates in terms of the cumulative reward during training. Our MASE obtains better policies in a smaller number of samples compared with Sauté RL, while also satisfying the safety constraints with respect to both the average and the worst-case. Note that, after convergence, the policy obtained by MASE performs worse than those obtained by the baseline algorithms in terms of reward, as shown in

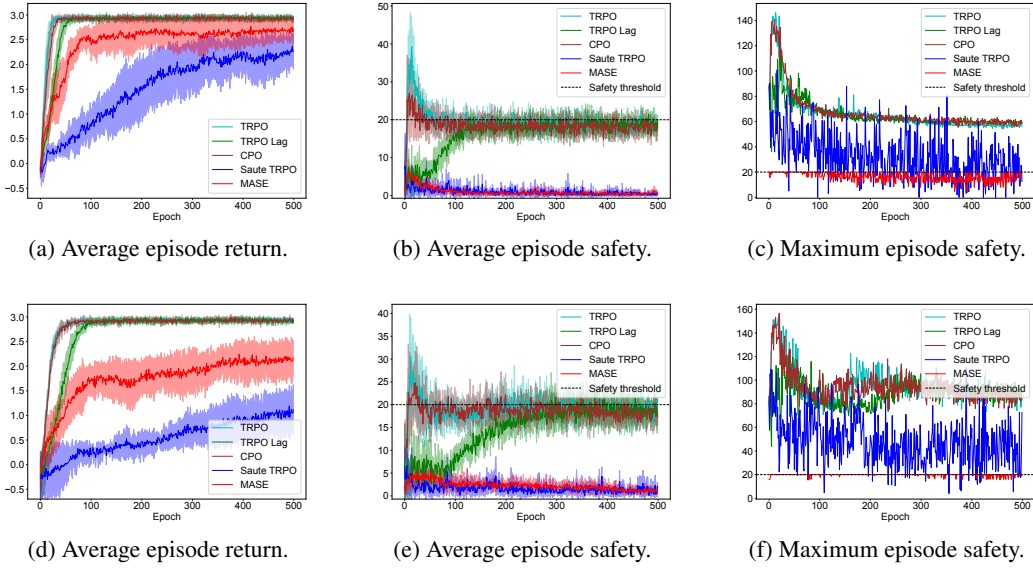

Figure 2: Experimental results on Safety Gym (Top: PointGoal1, Bottom: CarGoal1). The proposed MASE satisfies the safety constraint in every episode and achieves better performance in terms of the reward than the state-of-the-art method called Sauté RL. Conventional methods (i.e., TRPO, TRPO-Lagrangian, and CPO) repeatedly violate the safety constraint, especially in the early phase of training. Shaded areas represent $1\sigma$ confidence intervals across five different random seeds.

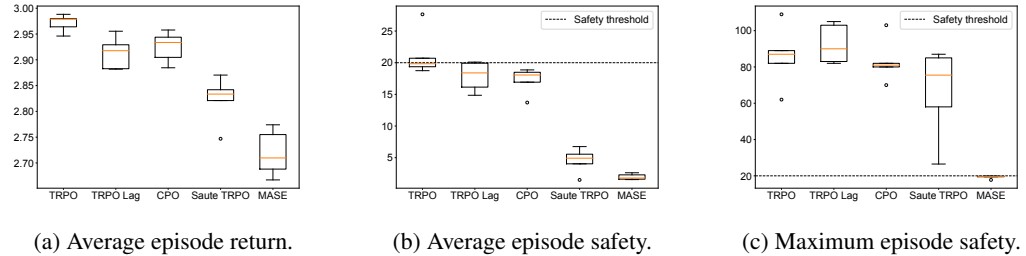

Figure 3: Experimental results on Safety Gym (CarGoal1) with the epoch of 1000. The box plots show the converged performance. Though MASE performs worse than other baselines in terms of reward, the acquired policy is still near-optimal. As for safety, while baselines violate the safety constraint in most of the episodes, MASE guarantees the satisfaction of the severe safety constraint.

Figure 3. The emergency stop action is a variant of resetting actions that are common in episodic RL settings, which prevent the agent from exploring the state-action spaces since the uncertainty quantifier is sometimes quite conservative. We consider that this is a reason why the converged reward performance of MASE is worse than other methods. However, because we require the agent to solve difficult problems where safety is guaranteed at every time step and episode, we consider that this result is reasonable, and further performance improvements are left to future work.

# 8 Conclusion

In this article, we first introduced the GSE problem and proved that it is more general than three common safe RL problems. We then proposed MASE to optimize a policy under safety constraints that allow the agent to execute an emergency stop action at the sacrifice of a penalty based on the $\delta$-uncertainty qualifier. As a specific instance of MASE, we first presented GLM-MASE to theoretically guarantee the near-optimality and safety of the acquired policy under generalized linear CMDP assumptions. Finally, we provided a practical MASE and empirically evaluated its performance in comparison with several baselines on the Safety Gym and grid-world.

## Acknowledgments and Disclosure of Funding

We would like to thank the anonymous reviewers for their helpful comments. This work is partially supported by JST CREST JPMJCR201 and by JSPS KAKENHI Grant 21K14184.

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

# Appendices

## A   Limitations and Potential Negative Societal Impacts

We will first discuss limitations and potential negative societal impacts regarding our work.

### A.1   Limitations

One of the major limitations of this study is that emergency stop actions are allowed for agents. Emergency stop actions should often be avoided because of the expensive cost of human intervention in many applications (e.g., the agent is in a hazardous or remote environment). In future work, we will investigate an algorithm that requires the agent to learn a reset policy allowing them to return to the initial state as in [15], rather than asking for human intervention via emergency stop actions.

Another limitation is how to construct the uncertainty quantifier. In our experiment, because we used a computationally inexpensive deep GP algorithm [30] and the uncertainty quantifier is updated at the end of the episode (see Line 13 in Algorithm 1), the computational time of the GP part was much smaller than the RL part in our experiment settings. However, since GP is generally a computationally expensive algorithm, GP can be a computational bottleneck in some cases.

### A.2   Potential Negative Societal Impacts

We believe that safety is an essential requirement for applying RL in many real problems. While we have not found any potential negative societal impact of our proposed method, we must remain aware that RL algorithms are vulnerable to misuse and ours is no exception.

## B   Proof of Theorem 3.1

We first present lemmas regarding the relationship between the GSE problem and Problems 1, 2, and 3. After that, we present the proof for the Theorem 3.1 in Appendix B.4 by combining them.

### B.1   Relationship between the GSE problem and Problem 1

**Lemma B.1.** *Problem 1 can be transformed into the GSE problem.*

*Proof.* We first utilize a safety state augmentation technique presented in Sootla et al. [32] by defining a new variable $\eta_h$ such that

$$\eta_h := \gamma_g^{-h} \cdot \left( \xi_1 - \sum_{h'=1}^{h-1} \gamma_g^{h'} g(s_{h'}, a_{h'}) \right), \quad \forall h \in [1, H]. \tag{4}$$

This new variable $\eta_h$ means the remaining safety budget associated with the discount factor $\gamma_g$, which is updated as follows:

$$\eta_{h+1} = \gamma_g^{-1} \cdot ( \eta_h - g(s_h, a_h) ) \quad \text{with} \quad \eta_0 = \xi_1. \tag{5}$$

By (4), the necessary and sufficient condition for satisfying the constraint in Problem 1 is

$$\eta_h \geq 0, \quad \forall h \in [1, H]. \tag{6}$$

By (5), we have

$$\eta_{h+1} \geq 0, \quad \forall h \in [1, H] \iff \eta_h - g(s_h, a_h) \geq 0, \quad \forall h \in [1, H]. \tag{7}$$

In summary, by introducing the new variable $\eta_h$, Problem 1 is rewritten to the following problem:

$$\max_{\pi} V_r^{\pi} \quad \text{subject to} \quad \Pr[ g(s_h, a_h) \leq \eta_h \mid \mathcal{P}, \pi ] = 1, \quad \forall h \in [1, H]. \tag{8}$$

The aforementioned problem (8) is a special case of the GSE problem with $b_h := \eta_h$. Therefore, we obtain the desired lemma. ☐

## B.2 Relationship between the GSE problem and Problem 2

**Lemma B.2.** *Problem 2 can be transformed into the GSE problem.*

*Proof.* The following chain of inequalities holds:

$$\mathbb{E}\left[\sum_{h=1}^{H}\gamma_g^h\,\mathbb{I}(s_h \in S_{\text{unsafe}})\,\Bigg|\,\mathcal{P},\pi\right] \leq \xi_2$$

$$\Longleftrightarrow \mathbb{E}\left[\sum_{h'=1}^{h}\gamma_g^{h'}\,\mathbb{I}(s_{h'} \in S_{\text{unsafe}})\,\Bigg|\,\mathcal{P},\pi\right] \leq \xi_2, \quad \forall h \in [1,H]$$

$$\Longleftrightarrow \mathbb{E}\left[\gamma_g^h\,\mathbb{I}(s_h \in S_{\text{unsafe}})\,\big|\,\mathcal{P},\pi\right] \leq \xi_2 - \mathbb{E}\left[\sum_{h'=1}^{h-1}\gamma_g^{h'}\,\mathbb{I}(s_{h'} \in S_{\text{unsafe}})\,\Bigg|\,\mathcal{P},\pi\right], \quad \forall h \in [2,H]$$

$$\Longleftrightarrow \mathbb{E}\left[\mathbb{I}(s_h \in S_{\text{unsafe}})\,|\,\mathcal{P},\pi\right] \leq \gamma_g^{-h}\left\{\xi_2 - \mathbb{E}\left[\sum_{h'=1}^{h-1}\gamma_g^{h'}\,\mathbb{I}(s_{h'} \in S_{\text{unsafe}})\,\Bigg|\,\mathcal{P},\pi\right]\right\}, \quad \forall h \in [2,H]$$

Because we assume Markov property, we simply define the safety cost function $g : \mathcal{S} \times \mathcal{A} \to \mathbb{R}$ as

$$g(s_h, a) := \mathbb{E}[\,\mathbb{I}(s_h \in S_{\text{unsafe}})\,|\,\mathcal{P},\pi\,], \quad \forall a \in \mathcal{A}.$$

We now set

$$b_h := \gamma_g^{-h}\left\{\xi_2 - \mathbb{E}\left[\sum_{h'=1}^{h-1}\gamma_g^{h'}\,\mathbb{I}(s_{h'} \in S_{\text{unsafe}})\,\Bigg|\,\mathcal{P},\pi\right]\right\},$$

then the Problem 2 can be transformed into

$$\Pr[\,g(s,a) \leq b_h\,|\,\mathcal{P},\pi\,] = 1.$$

Finally, we obtained the desired lemma. $\qquad\square$

## B.3 Relationship between the GSE problem and Problem 3

**Lemma B.3.** *Problem 3 can be transformed into the GSE problem.*

*Proof.* Set $b_h = \xi_3$ for all $h$. Then, the GSE problem is identical to Problem 3. $\qquad\square$

## B.4 Summary

*Proof.* Combining Lemma B.1, B.2, and B.3, we obtain the desired Theorem 3.1. $\qquad\square$

# C Connections to Safe RL Problems with Chance Constraints

As a strongly related formulation to Problem 2, policy optimization under joint chance-constraints has been studied especially in the field of control theory such as Ono et al. [24] and Pfrommer et al. [26], which is written as

**Problem 5** (Safe RL with joint chance constraints)**.** Let $\xi_5 \in \mathbb{R}_{\geq 0}$ be a constant representing a safety threshold. Also, let $S_{\text{unsafe}} \subset \mathcal{S}$ denote a set of unsafe states. Find the optimal policy $\pi^\star$ such that

$$\max_\pi V_r^\pi \quad \text{subject to} \quad \Pr\left[\bigvee_{h=1}^{H} s_h \in S_{\text{unsafe}}\,\Bigg|\,\mathcal{P},\pi\right] \leq \xi_5.$$

**Lemma C.1.** *Problem 2 is a conservative approximation of Problem 5.*

*Proof.* This lemma mostly follows from Theorem 1 in Ono et al. [24]. Regarding the constraint in Problem 5, we have the following chain of equations:

$$\Pr\left[\bigvee_{h=1}^{H} s_h \in S_{\text{unsafe}} \,\middle|\, \mathcal{P}, \pi\right] \leq \sum_{h=1}^{H} \Pr\left[s_h \in S_{\text{unsafe}} \mid \mathcal{P}, \pi\right]$$

$$= \sum_{h=1}^{H} \mathbb{E}\left[\mathbb{I}(s_h \in S_{\text{unsafe}}) \mid \mathcal{P}, \pi\right] \tag{9}$$

$$= \mathbb{E}\left[\sum_{h=1}^{H} \mathbb{I}(s_h \in S_{\text{unsafe}}) \,\middle|\, \mathcal{P}, \pi\right]. \tag{10}$$

In the first step, we used Boole's inequality (i.e., $\Pr[A \cup B] \leq \Pr[A] + \Pr[B]$). The final term in (10) is the LHS of the constraint in Problem 2, which implies that Problem 2 is a conservative approximation of Problem 5. Therefore, the GSE problem is also a conservative approximation of Problem 5. $\square$

**Corollary C.2.** *Suppose we solve the GSE problem by properly defining the safety function $g(\cdot, \cdot)$ and the threshold $b_h$. Then, the obtained policy is a conservative solution of Problem 5.*

**Remark C.3.** It is extremely challenging to directly solve the Problem 5 characterized by *joint* chance-constraints without approximation, as discussed in Ono et al. [24]. Practically, it would be a promising and realistic approach to solve Problem 5 by converting it into the GSE problem.

More detailed explanations for the aforementioned remark are as follows. It is extremely challenging to directly solve the Problem 5 characterized by *joint* chance-constraints. Most of the previous work does not directly deal with this type of constraint and uses some approximations or assumptions. For example, Pfrommer et al. [26] assume a known linear time-invariant dynamics. Also, Ono et al. [24] approximate the joint chance constraint as in the above procedure and obtain

$$\Pr\left[\bigvee_{h=1}^{H} s_h \in S_{\text{unsafe}} \,\middle|\, \mathcal{P}, \pi\right] \leq \mathbb{E}\left[\sum_{h=1}^{H} \mathbb{I}(s_h \in S_{\text{unsafe}}) \,\middle|\, \mathcal{P}, \pi\right].$$

This is a conservative approximation with an additive structure, which is easier to solve than the original joint chance constraint. Ono et al. [24] deals with the above constraints with additive safety structure. By additionally transforming the conservatively-approximated problem into the GSE problem, the problem would become easier to handle because the safety constraint is instantaneous.

# D    Proof of Theorems 4.1 and 4.2

## D.1    Proof of Theorem 4.1

*Proof.* Recall that, at every time step $h$, the MASE chooses actions satisfying

$$\mu(s_h, a_h) + \Gamma(s_h, a_h) \leq b_h. \tag{11}$$

By definition of the $\delta$-uncertainty quantifier, the actions that are conservatively chosen based on (11) also satisfy the safety constraint, with a probability of at least $1 - \delta$; that is,

$$g(s_h, a_h) \leq b_h. \tag{12}$$

In addition, when there is no action satisfying (11), the emergency stop action is executed; that is, safety is guaranteed with a probability of 1. Hence, the desired theorem is now obtained. $\square$

## D.2    Proof of Theorem 4.2

*Proof.* Assumption 3.2 implies that there exists a policy that satisfies a more conservative safety constraint written as

$$g(s_h, a_h) \leq b_h - \zeta, \quad \forall h \in [1, H]. \tag{13}$$

By combining (13) and the assumption $\Gamma(s, a) \leq \frac{1}{2}\zeta, \forall(s, a) \in \mathcal{S} \times \mathcal{A}$, we have

$$\mu(s_h, a_h) + \Gamma(s_h, a_h) \leq g(s_h, a_h) + \zeta \leq b_h, \quad \forall h \in [1, H], \tag{14}$$

---

**Algorithm 2** GLM-MASE

---

1: **for** episode $t = 1, 2, \cdots, T$ **do**
2:      **for** time $h = 1, \ldots, H$ **do**
3:          Take action $a_h = \pi(s_h)$ within $\mathcal{A}_h^+$
4:          Receive reward $r(s_h, a_h)$ and next state $s_{h+1}$
5:          Receive safety cost $g(s_h, a_h)$ and update safety threshold $b_{h+1}$
6:          **if** $\mathcal{A}_h^+ = \emptyset$ **then**
7:              Compute $\widehat{r}(s_h, a_h) = -\frac{c}{\min_{a \in \mathcal{A}} \Gamma(s_{h+1}, a)}$
8:              Append $(s_h, a_h, \widehat{r}(s_h, a_h), s_{h+1})$ to $\mathcal{D}$
9:              **break** (i.e., emergency stop action $\widehat{a}$)
10:          **else**
11:              Append $(s_h, a_h, r(s_h, a_h), s_{h+1})$ to $\mathcal{D}$
12:      Update $\widehat{\theta}_{h,t}^g$ and $\widehat{\theta}_{h,t}^Q$
13:      Compute the optimistic Q-estimate

$$\widehat{Q}_{\widehat{r},h}^{(t)}(s, a) = \min\{V_{\max}, f(\langle \phi_{s,a}, \hat{\theta}_{h,t}^Q \rangle) + C_{Q/g} \Gamma(s, a)\}$$

14:      Optimize the policy by

$$\pi_h^{(t)}(s) = \arg\max_{a \in \mathcal{A}} \widehat{Q}_{\widehat{r},h}^{(t)}(s, a).$$

15:      Update $\Gamma(s, a) := C_g \cdot \|\phi_{s,a}\|_{\Lambda_{h,t}^{-1}}$ and then rewrite $\mathcal{D}$

---

which guarantees that there exists a policy that conservatively satisfies the safety constraint via the $\delta$-uncertainty quantifier $\Gamma(\cdot, \cdot)$ at every time step $h$, with a probability of at least $1 - \delta$.

When we set $c$ to be a sufficiently large scalar such that $c > \frac{\zeta}{2\gamma_r^H} V_{\max}$, the penalty $\widehat{r}$ satisfies

$$\begin{aligned}
\widehat{r}(s_h, a_h) &= \frac{-c}{\min_{a \in \mathcal{A}} \Gamma(s_{h+1}, a)} \\
&< \frac{-\frac{1}{2}\zeta \cdot \frac{1}{\gamma_r^H} V_{\max}}{\frac{1}{2}\zeta} \\
&< -\gamma_r^{-H} V_{\max}.
\end{aligned}$$

This means that, when the constraint violation happens even a single time, the value by a policy obtained in $\widehat{\mathcal{M}}$ becomes negative because $\max_s V_r^\pi(s) \leq V_{\max}$.

Under Assumption 3.2, after convergence, the optimal policy in $\widehat{M}$ will not violate the safety constraint, and thus the emergency stop action $\widehat{a}$ will not be executed. In this case, the modified (unconstrained) MDP $\widehat{\mathcal{M}}$ is identical to the original CMDP $\mathcal{M}$. Therefore, we now obtain the desired theorem. $\qquad\square$

# E    Supplementary materials regarding GLM-MASE

## E.1    Pseudo-code for GLM-MASE

We first present the pseudo-code for GLM-MASE in Algorithm 2.

## E.2    Proofs of Lemmas 5.4 and 5.5

*Proof.* See Lemma 1 (and Lemma 7) in Wang et al. [44]. $\qquad\square$

## E.3    Preliminary Lemmas

**Lemma E.1.** *Suppose the assumptions in Lemma 5.4 and 5.5 hold. Let $C_1$ and $C_2$ be positive, universal constants. Also, with a sufficiently large $T$, let $t^\star$ denote the smallest integer satisfying*

$$\lambda_{\min}(\Sigma)tH - C_1\sqrt{tHd} - C_2\sqrt{tH \ln \delta^{-1}} \geq 2 \cdot C_g \cdot \zeta^{-1} \tag{15}$$

where $\lambda_{\min}(\Sigma)$ is the minimum eigenvalue of the second moment matrix $\Sigma$. Then, we have

$$\Gamma(s_h^{(t)}, a_h^{(t)}) \leq \frac{1}{2} \cdot \zeta. \tag{16}$$

*Proof.* By Proposition 1 of Li et al. [22],

$$\lambda_{\min}(\Lambda_{h,t}) \geq \lambda_{\min}(\Sigma)tH - C_1\sqrt{tHd} - C_2\sqrt{tH\ln\delta^{-1}}.$$

By combining the aforementioned inequality with (15), we have

$$\lambda_{\min}(\Lambda_{h,t}) \geq 2 \cdot C_g \cdot \zeta^{-1}.$$

Using the definition of $\lambda_{\max}(\Lambda_{h,t}^{-1}) = \frac{1}{\lambda_{\min}(\Lambda_{h,t})}$, the following chain of equations hold for all $t \in [t^\star, T]$ and $h \in [1, H]$:

$$\begin{aligned}
\Gamma(s_h^{(t)}, a_h^{(t)}) &= C_g \cdot \|\phi_{s,a}\|_{\Lambda_{h,t}^{-1}} \\
&\leq C_g \cdot \lambda_{\max}(\Lambda_{h,t}^{-1}) \\
&= C_g \cdot \lambda_{\min}^{-1}(\Lambda_{h,t}) \\
&\leq \frac{1}{2}\zeta.
\end{aligned}$$

Therefore, we have the desired lemma. $\square$

### E.4 Proof of Theorem 5.6

*Proof.* By definition, the GLM-MASE chooses actions satisfying

$$f(\langle \widetilde{\phi}_h^{(\tau)}, \widehat{\theta}_{h,t}^g \rangle) + \Gamma(s_h^{(t)}, a_h^{(t)}) \leq b_h. \tag{17}$$

By Lemma 5.4, the actions that are conservatively chosen based on (17) also satisfy the actual safety constraint, with a probability at least $1 - \delta$; that is,

$$g(s_h^{(t)}, a_h^{(t)}) \leq b_h. \tag{18}$$

In addition, when there is no action satisfying (17), the emergency stop action is executed where no unsafe action will be executed. Hence, the desired theorem is now obtained. $\square$

### E.5 Proof of Theorem 5.7

By Assumption 3.2, the optimal policy $\pi^\star$ satisfies

$$g(s_h, \pi^\star(s_h)) \leq b_h - \zeta, \quad \forall h \in [1, H]. \tag{19}$$

Thus, the set of state-action pairs that are potentially visited by $\pi^\star$ are written as

$$\{(s, a) \in \mathcal{S} \times \mathcal{A} \mid g(s, a) \leq b_h - \zeta\},$$

which satisfies the following chain of inequalities:

$$\begin{aligned}
\{(s, a) \mid g(s, a) \leq b_h - \zeta\} &\subseteq \{(s, a) \mid \mu(s, a) - \Gamma(s, a) \leq b_h - \zeta\} \\
&\subseteq \{(s, a) \mid \mu(s, a) + \Gamma(s, a) \leq b_h\}.
\end{aligned}$$

The state and action spaces in the last line represent the set of state-action pairs that may be visited by the policy obtained by the MASE algorithm. We used Lemma 5.4 in the first line and $\Gamma(s, a) < \frac{\zeta}{2}$ in the second line.

By Lemma 8 in Strehl and Littman [34], the total regret can be decomposed into two parts as follows:

$$\sum_{t=t^\star}^{T} \left[ V_r^{\pi^\star} - V_r^{\pi_t} \right] = \mathcal{R}(T) + \sum_{t=t^\star} V_{\max}\delta, \tag{20}$$

where the first term is the regret under the condition that the confidence bound based on $\delta$-uncertainty quantifier successfully contains the true safety function. Also, the second term is the regret under the opposite condition, which occurs with a probability $\delta$.

The first term in (20) is upper-bounded based on Wang et al. [44] as follows:

$$\mathcal{R}(T) \leq O\Bigg( H\sqrt{(T - t^\star)\ln((T - t^\star)H)}$$

$$+ H\overline{\kappa}\underline{\kappa}^{-1}\sqrt{M + \overline{\kappa} + d^2 \ln\left(\frac{\overline{\kappa} + \alpha_{\max}}{(T - t^\star)H}\right) \cdot (T - t^\star)d\ln\left(1 + \frac{(T - t^\star)}{d}\right)}\Bigg)$$

$$\leq \widetilde{O}(H\sqrt{d^3(T - t^\star)}).$$

As for the second term in (20), set $\delta = \frac{1}{TH}$ and then we have

$$\sum_{t=t^\star}^{T} V_{\max} \cdot \delta = \sum_{t=t^\star}^{T} V_{\max} \cdot \frac{1}{TH}$$

$$= \sum_{t=t^\star}^{T} \frac{1 - \gamma^H}{1 - \gamma} \cdot \frac{1}{TH}$$

$$\leq \sum_{t=t^\star}^{T} H \cdot \frac{1}{TH}$$

$$\leq O(1).$$

In summary, the regret can be upper bounded by $\widetilde{O}(H\sqrt{d^3(T - t^\star)})$. We now obtain the desired theorem.

## F    Proof of Theorem 6.1

**Lemma F.1** ($\delta$-uncertainty quantifier)**.** *Assume* $\|g\|_k^2 \leq B$ *and* $N_n \leq \omega$ *for all* $n \geq 1$*. Set*

$$\Gamma(s, a) := \beta_n \cdot \sigma_n(s, a), \quad \forall(s, a) \in \mathcal{S} \times \mathcal{A} \tag{21}$$

*with* $\beta_n^{1/2} := B + 4\omega\sqrt{\nu_n + 1 + \ln \delta^{-1}}$*, where* $\nu_n$ *is the information capacity associated with kernel* $k$*. Then,* $\Gamma$ *is a* $\delta$*-uncertainty quantifier.*

*Proof.* Recall the assumption that $\|g\|_k^2 \leq B$ and $N_n \leq \omega, \forall n \geq 1$. Also, set

$$\beta_n^{1/2} := B + 4\omega\sqrt{\nu_n + 1 + \ln \delta^{-1}}.$$

By Theorem 2 in Chowdhury and Gopalan [12], we have

$$|g(s, a) - \mu_n(s, a)| \leq \beta_n \cdot \sigma_n(s, a), \quad \forall(s, a) \in \mathcal{S} \times \mathcal{A}$$

for all $n \geq 1$, with probability at least $1 - \delta$. Now, define

$$\Gamma(s, a) := \beta_n \cdot \sigma_n(s, a), \quad \forall(s, a) \in \mathcal{S} \times \mathcal{A} \tag{22}$$

then we interpret that $\Gamma : \mathcal{S} \times \mathcal{A} \to \mathbb{R}$ is a $\delta$-uncertainty quantifier based on GP.  $\square$

### F.1    Proof of Theorem 6.1

*Proof.* Based on Lemma F.1, when there is at least one safe action (i.e., $\mathcal{A}_h^+ \neq \emptyset$), the satisfaction of the safety constraint is guaranteed based on the $\delta$-uncertainty quantifier with a probability at least $1 - \delta$. Also, if there is no safe action (i.e., $\mathcal{A}_h^+ = \emptyset$), the emergency stop action is executed. In both cases, MASE guarantees the satisfaction of the safety constraint with probability at least $1 - \delta$.  $\square$

Table 1: Hyper-parameters for Safety Gym experiments.

| | NAME | VALUE |
|---|---|---|
| | NETWORK ARCHITECTURE | $[64, 64]$ |
| | ACTIVATION FUNCTION | tanh |
| | LEARNING RATE (CRITIC) | $5 \times 10^{-3}$ |
| | LEARNING RATE (POLICY) | $3 \times 10^{-4}$ |
| | LEARNING RATE (PENALTY) | $3 \times 10^{-2}$ |
| COMMON PARAMETERS | DISCOUNT FACTOR (REWARD) | 0.99 |
| | DISCOUNT FACTOR (SAFETY) | 0.99 |
| | STEPS PER EPOCH | $10,000$ |
| | NUMBER OF GRADIENT STEPS | 80 |
| | NUMBER OF EPOCHS | 500 |
| | TARGET KL | 0.01 |
| | DAMPING COEFFICIENT | 0.1 |
| TRPO & CPO | BACKTRACK COEFFICIENT | 0.8 |
| | BACKTRACK ITERATIONS | 10 |
| | LEARNING MARGIN | FALSE |
| | PENALTY FOR EMERGENCY STOP ACTIONS | $-1$ |
| MASE | DEEP GP NETWORK ARCHITECTURE | $[16, 16]$ |
| | NUMBER OF INDUCING POINTS | 128 |
| | KERNEL FUNCTION | RADIAL BASIS FUNCTION |

## G  Details of Safety-Gym Experiment

We present the details regarding our experiments using Safety-Gym. Our experimental setting is based on Sootla et al. [32], which is slightly different from the original Safety Gym in that the obstacles (i.e., unsafe region) are replaced deterministically so that the environment is solvable and there is a viable solution. In this experiment, we used a machine with Intel(R) Xeon(R) Silver 4210 CPU, 128GB RAM, and NVIDIA A100 GPU. For a fair comparison, we basically used the same hyper-parameter as in Sootla et al. [32], which is summarized in Table 1.

In our experiment, when the agent identified that there was no safe action based on the GP-based uncertainty quantifier, we simply terminated the current episode (i.e., resetting) immediately after the emergency stop action and started the new episode. The frequency of the emergency stop actions is shown in Table 2.

Table 2: Frequency of the emergency stop actions.

| TASK | TOTAL | LAST 100 EPISODES |
|---|---|---|
| POINTGOAL1 | 154/500 | 24/100 |
| CARGOAL1 | 397/500 | 46/100 |

The emergency stop action is a variant of so-called resetting actions that are common in episodic RL settings, which prevent the agent from exploring the state-action spaces since the uncertainty quantifier is sometimes quite conservative. We consider that this is the reason why the reward performance of our MASE is worse than other methods (e.g., TRPO-Lagrangian, CPO). However, because we require the agent to solve more difficult problems where safety is guaranteed at every time step and episode, we consider that this result is reasonable to some extent. Though it is better for an algorithm for such a severe safety constraint to have a comparable performance as CPO, we will leave it for future work.

We also conducted an experiment to compare the performance of MASE with the early-terminated MDP (ET-MDP, [37]) algorithm. The ET-MDP is an algorithm to execute emergency stop actions *immediately after* safety constraints are violated. Figure 4 shows the experimental results. The ET-MDP and MASE exhibit similar learning curves on the average episode reward and average episode safety. However, while ET-MDP violated the safety constraint in most episodes (i.e., almost

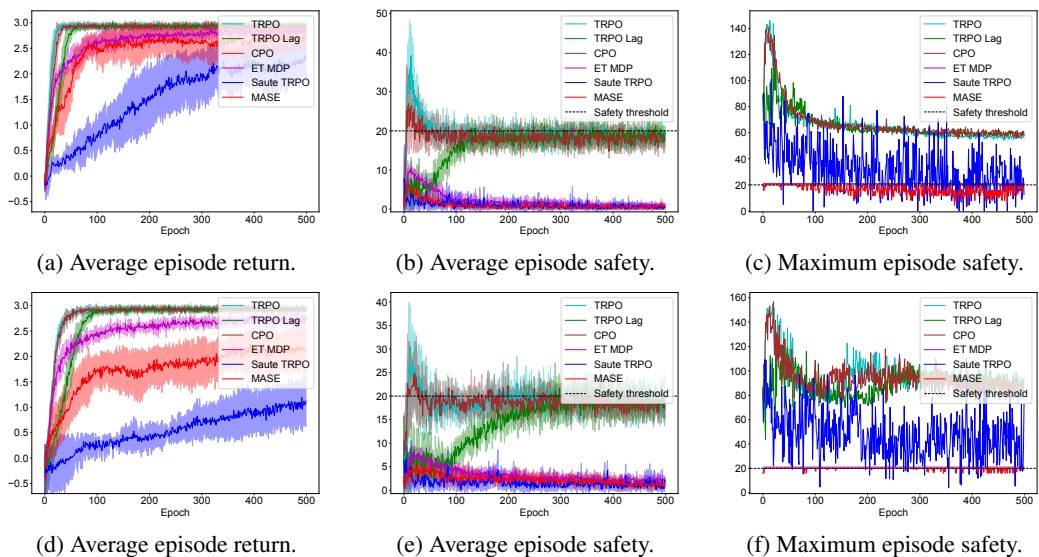

Figure 4: Experimental results on Safety Gym (Top: PointGoal1, Bottom: CarGoal1) with an additional implementation of the Early-terminated MDP (ET-MDP) algorithm (Sun et al. [37]).

all episodes are terminated after an unsafe action is executed), MASE did not violate any safety constraint.

# H  Grid-world Experiment

We also conduct an experiment using the grid-world problem as in Wachi and Sui [41]. Experimental settings are based on their original implementation (`https://github.com/akifumi-wachi-4/safe_near_optimal_mdp`). We consider a $20 \times 20$ square grid in which reward and safety functions are randomly generated. There are two types regarding the safety threshold: one is time-invariant as in [41] and the other is time-variant as in the GSE problem.

We run SNO-MDP [41] and MASE in $100$ randomly generated environments, and we compute the reward collected by the algorithms and count the number of episodes in which the safety constraint is violated at least once. The reward is normalized with respect to that by SNO-MDP.

Table 3: Experimental results for grid-world experiments.

|  | TIME-INVARIANT SAFETY THRESHOLD | | TIME-VARIANT SAFETY THRESHOLD | |
| --- | --- | --- | --- | --- |
|  | REWARD | SAFETY VIOLATION | REWARD | SAFETY VIOLATION |
| SNO-MDP [41] | $1.0 \pm 0.0$ | 0 | $1.0 \pm 0.0$ | 87 |
| MASE (OURS) | $1.0 \pm 0.0$ | 0 | $2.4 \pm 1.0$ | 0 |

The experimental results are shown in Table 3. When the safety threshold is time-invariant, MASE behaves identically with SNO-MDP; thus, the performance of the MASE is comparable with that of SNO-MDP. When the safety threshold is time-variant, SNO-MDP cannot deal with it by nature; hence, the safety constraint is not satisfied in most of the episodes. In contrast, our MASE satisfies the safety constraint in every episode, which also contributes to the larger reward.

