# OpenReview forum: "Safe Exploration in Reinforcement Learning: A Generalized Formulation and Algorithms"
_NeurIPS.cc/2023/Conference — NeurIPS 2023 poster_

### Official Review · Reviewer_pGdw · 2023-06-24

**Soundness:** 3 good
**Presentation:** 3 good
**Contribution:** 3 good
**Rating:** 7
**Confidence:** 4

**Summary:**

This paper studies the safe RL problem with a generalized stepwise safety chance (probability) constraint, essential for many safety-critical systems with RL. The authors propose a meta-algorithm to solve this problem (MASE), by combining unconstrained RL with an uncertainty quantifier to guarantee safety with probability. They study two variants for MASE, one for the linear model and the other for GP. Experiments on grid-world and safety gym show that MASE with GSE formulation achieves SOTA compared to the baselines.

**Strengths:**

The paper is well-written and easy to follow. The proposed GSE problem is more general than the CMDP formulation with an additive expectation constraint. Especially, the GSE problem is an important problem to study for many real-world safety-critical systems, such as autonomous cars and cars.
The proposed method looks sound and correct to me.
The MASE algorithm essentially builds on Assumption 3.4 (uncertainty quantifier) which could also be a potential bottleneck or limitation, but the author did a good job by proposing a general linear model and GP for it.
The experimental results show better results in terms of safety violations, compared to other CMDP-based approaches.

**Weaknesses:**

1. The author may want to at least discuss this ICML paper in their related work. The hard safety chance constraint is highly relevant to the GSE problem in this paper, although it is indeed using different approaches to solve the problem.
Wang, Y., Zhan, S. S., Jiao, R., Wang, Z., Jin, W., Yang, Z., ... & Zhu, Q. (2022). Enforcing Hard Constraints with Soft Barriers: Safe Reinforcement Learning in Unknown Stochastic Environments. arXiv preprint arXiv:2209.15090.

2. The authors should discuss the limitations of this work

3. In Method, MASE needs to compute a safe action set, how complex this computation is? In order to compute it, what kind of assumptions of the underlying environments do you acquire?

**Questions:**

1. Page 3 Lines 96 to 98, although I can understand what the authors mean, we should always keep in mind that s_h (a_h) are random variables, you cannot let a random variable be less than a value. It has to be in Pr() format, even if the probability is 1.

**Limitations:**

I would like to see the authors' opinions on the limitations in their responses.

---

> ### Author Rebuttal · Authors · 2023-08-08
>
> We deeply appreciate the reviewer’s encouraging comments. Your feedback and suggestions are valuable and help us improve the quality of the manuscript.
>
> We will first answer the Question and then address the comments regarding Weakness.
>
> **Lines 96 - 98 [Question].** Thank you for the important question! As the reviewer mentions, we should have paid more attention to the fact that $s_h$ and $a_h$ are random variables. We will fix them in the camera-ready version.
>
> **Missing reference [Weakness 1].** Thank you for pointing out the closely-related work. Indeed, the existing work [Ref-1] the reviewer points out formulates their problem in a highly-relevant manner to the GSE problem. We will discuss this existing work as a closely-related research.
>
> **Limitations of the work [Weakness 2].** Due to the severe page limit, we discuss the limitations of our paper in the supplementary material (Appendix I). As the reviewer mentioned, however, the limitations should be discussed in the main paper and it would be better to discuss other limitations such as computational cost. We will move this part to the main paper in the camera-ready version while adding more descriptions.
>
> **Complexity [Weakness 3].** In our proposed method, a safe action set is computed based on the inference by uncertainty quantifier (e.g., GP). In our practical implementation, since each GP inference is computationally inexpensive and we require the agent to update the GP model only at the end of the episode (see Line 13 in Algorithm 1), dominant computational time depends on the number of actions used for the GP inference. In the case of RL problems with discrete action, the computational cost is proportional to $|A|$. When the action space is continuous, it would be more difficult to compute the safe action set and a sampling technique is a simple yet powerful solution. In fact, when we conducted our experiment, we randomly sampled next actions and checked whether or not there were actions that conservatively guarantee the safety constraint. The computational cost of this process is practically smaller than the main RL process. We should have explained the details of the practical implementation, so we will address the reviewers’ comments in the next version.
>
> We sincerely thank the reviewer for taking the time to review our paper.
>
> ---
> Reference
>
> [Ref-1] Wang, Yixuan, et al. "Enforcing hard constraints with soft barriers: Safe reinforcement learning in unknown stochastic environments." International Conference on Machine Learning (ICML)., 2023.

---

### Official Review · Reviewer_zpL1 · 2023-07-01

**Soundness:** 3 good
**Presentation:** 2 fair
**Contribution:** 3 good
**Rating:** 7
**Confidence:** 4

**Summary:**

The authors present a novel safe RL algorithm for safety constraints with probability one. First, the authors present a problem formulation that can be used to derive different common safe RL formulation (state constraints, accumulated constraints etc). Then the authors leverage a sophisticated technique to reshape the reward accounting for safety information. In particular, they predict if the following actions will be safe by learning the action safety state similar to safety shield techniques. Their shields are learning using Gaussian processes and offer safety predictions with uncertainty estimates.`


**Strengths:**

1. The algorithm is technically sound, novel and quite interesting as an idea.
1. The authors present an extensive analysis for the case of generalized linear CMPDs. The theoretical results seem to be correct, but I have a few minor concerns regarding the statements
2. The numerical results are impressive but shown only for one environment.


**Weaknesses:**

1. The statement of Theorem 3.1 reads there are instances of the GSE problem that are not equivalent and cannot be transformed into Problem 1, 2 or 3. This is not shown.
1. The fact that Problem 2 can be transformed into the GSE problem does not mean the GSE problem is more general than Problem 2. If the GSE problem can be transformed to Problem 2 as well, then they are equivalent. Furthermore, if the accumulated cost is used as a state as in Lemma A.1, then the problems cannot truly be equivalent. I think the authors should rephrase these results and make more accurate claims.
1. The method does remind me of safety layer techniques [Dalal 18] and [28] with a more sophisticated reward-shaping approach. Can the authors provide a short discussion on the relation to these papers?
1. The numerical results can be improved
     * I recommend adding boxplots to the simulation results to see the distributions of the traces. For instance the boxplots of the trajectories for the final epoch as in [28]
     * I recommend providing more epochs in the experiments. I suspect that the algorithms generally achieve similar performance, but the algorithms with probability one constraint simply converge slower.

* [Dalal 18] Dalal, G., Dvijotham, K., Vecerik, M., Hester, T., Paduraru, C., & Tassa, Y. (2018). Safe exploration in continuous action spaces. arXiv preprint arXiv:1801.08757.
* [Yu 22] Yu, Haonan, Wei Xu, and Haichao Zhang. "Towards safe reinforcement learning with a safety editor policy." Advances in Neural Information Processing Systems 35 (2022): 2608-2621.

**Questions:**

1. Please provide details on the safety gym environment. This does not seem like a standard one.
1. How the algorithm scales with a number of epochs/states? My concern is that the GP is not the most scalable model for RL


**Limitations:**

There are two limitations of the approach: 1) the scalability of the algorithm and 2) the restrictiveness of the problem definition. While the authors discuss the latter limitation and show that this problem formulation is important, I didn’t find a discussion on scalability.

---

> ### Author Rebuttal · Authors · 2023-08-08
>
> We deeply appreciate the reviewer’s encouraging comments. Your thoughtful comments are valuable and help us to improve the quality of the manuscript.
>
> We will answer the Questions first and then address the comments in Weakness.
>
> **Safety-Gym [Question 1].** Thank you for the very important question! Our experimental setting is based on the SOTA paper [28]. The Safety-gym environment is slightly different from the original one in that the obstacles (i.e., unsafe region) are replaced deterministically so that the environment is solvable and there is a viable solution. Because this information is important for reproducing our work, we will add the above explanations in the camera-ready version.
>
> **Scalability [Question 2].** Thank you for the valuable question! As the reviewer mentions, in general, GP is a computationally-expensive algorithm. However, computationally scalable GP algorithms have been proposed such as [Ref-1] or [Ref-2]. For example, KISS-GP (w/ LOVE) [Ref-1] requires $\mathcal{O}(k(n + m \log m))$, where $n$ is training points, $m$ inducing points, $k$ Lanczos/CG iterations. In our experiment, we used a scalable deep GP algorithm [26] and the computational time of the GP part was much smaller than the RL part in our experiment settings. This is also because of the algorithm setup that the uncertainty quantifier is updated at the end of the episode (see Line 13 in Algorithm 1). As the reviewer points out, since GP can be a computational bottleneck in some cases, we will discuss its potential issue in Limitation.
>
> **Terminology issue in Theorem 3.1 [Weakness 1 and Weakness 2].** We apologize that Theorem 3.1 is miss-leading. We provide Theorem 3.1 to insist that *"Problems 1, 2, and 3 can be transformed into the GSE problem."* Notice that the MASE algorithm can solve the GSE problem. By Theorem 3.1, we can further claim that the MASE algorithm can also solve Problems 1, 2, and 3 since they can be transformed into the GSE problem. Namely, Theorem 3.1 implies that the MASE algorithm is useful in many safe RL problems. The current statement is confusing and over-claiming more than necessary, which hides our real implications; hence, we will rephrase the statement around Theorem 3.1 and make more accurate claims. Thank you for your helpful comments.
>
> **Comparison with [Ref-3], [Ref-4], and [28] [Weakness 3].** As the reviewer mentions, [Ref-3] solves a similar problem to ours. The biggest difference between [Ref-3] and our paper is that [Ref-3] basically assumes that there is always at least one safe action at every time step while our MASE incorporates situations where there is no safe action. Because our MASE algorithm is developed for solving the GSE problem characterized by a time-varying (potentially decreasing) safety threshold $b_h$, we need to carefully consider the case where there is no safe action. The advantage of the MASE compared to [Ref-3] is that it can deal with such hopeless cases by the emergency stop action and reward penalty based on the uncertainty quantifier, while maintaining the theoretical guarantees on safety and (optimality under the Generalized linear CMDP assumption). Next, [Ref-4] tries to solve typical safe RL problems with expected cumulative safety constraints; thus, it is essentially difficult to guarantee the safety constraint in the GSE problem and Problems 1, 2, and 3. Finally, while [28] deals with the so-called probability one constraints (i.e., Problem 1 in our paper), their algorithm heavily penalizes the agent after the constraint violation. Their proposed algorithm is nice in that safety after convergence is guaranteed, but safety is not guaranteed during the learning phase by nature. An advantage of our MASE compared to theirs is that safety is guaranteed even during learning, which is evidenced by both theoretically and empirically.
>
> Note that, we thought the reviewer may have miswritten [Ref-4] as [28] and gave the above response.
>
> **Numerical results [Weakness 4].** Thank you for the useful comments. We conducted an additional experiment results with boxplots for a larger epoch (500 $\rightarrow$ 1000). **We would like to ask the reviewer to see the new result in a new PDF file attached in the global response.** Because we consider that it is more important to show the performance in terms of reward and safety during learning, we show the current learning curve. As the reviewer mentions, however, the boxplots would be definitely useful to see the final performance and we will add such plots as the new figure (i.e. Figure 4 in a PDF attached in the global response).
>
> We sincerely thank the reviewer for taking the time to review our paper.
>
> ---
> References
>
> [26] Salimbeni, H. and Deisenroth, M. Doubly stochastic variational inference for deep Gaussian processes. In Neural Information Processing Systems, 2017.
>
> [28] Sootla, Aivar, et al. "Sauté rl: Almost surely safe reinforcement learning using state augmentation." International Conference on Machine Learning. PMLR, 2022.
>
> [Ref-1] Wilson, A, and Hannes N. "Kernel interpolation for scalable structured Gaussian processes (KISS-GP)." International conference on machine learning. 2015.
>
> [Ref-2] Pleiss, G, et al. "Constant-time predictive distributions for Gaussian processes." International Conference on Machine Learning. 2018.
>
> [Ref-3] Dalal, G., Dvijotham, K., Vecerik, M., Hester, T., Paduraru, C., & Tassa, Y. (2018). Safe exploration in continuous action spaces. arXiv preprint arXiv:1801.08757.
>
> [Ref-4] Yu, Haonan, Wei Xu, and Haichao Zhang. "Towards safe reinforcement learning with a safety editor policy." Advances in Neural Information Processing Systems 35 (2022): 2608-2621

---

> > ### Comment · Reviewer_zpL1 · 2023-08-11
> > **Final remarks**
> >
> > I thank the authors for the responses and the new experiments. I have two remarks:
> >
> > **Terminology issue in Theorem 3.1 [Weakness 1 and Weakness 2].** I agree with the formulation that Problems 1,2, and 3 can be transformed to GSE for Theorem 3.1. I also suggest clarifying that GSE can be used to solve other problems, but there is no (to our best knowledge) direct proof that solving other problems would not solve GSE. Unless there is proof and that could be another interesting addition.
> >
> > **Comparison with [Ref-3], [Ref-4], and [28] [Weakness 3]** Just a minor remark that I think was alluded to in other responses. I believe your algorithm for probability one constraint can be seen as a nice generalization of [28] with (again) a nice form of shielding. This is naturally a matter of opinion, but I found this could be an interesting connection. If some connections can be made perhaps the theory of [28] can be extended to MASE.
> >
> > I also suggest adding boxplots to the final version of the paper not to the appendix. I feel this would be a fair comparison to [28] showing the superior performance of MASE.

---

> > > ### Author Response · Authors · 2023-08-12
> > > **Thank you for further comments.**
> > >
> > > We would like to express our sincere gratitude to the reviewer who read through our responses and other reviews.
> > >
> > > **Terminology issue in Theorem 3.1.** Thank you for your valuable comments. We will make sure that the terminology issues around Theorem 3.1 are fixed to avoid misleading and over-claiming statements. Also, thank you for additional suggestions regarding the theoretical analysis of the GSE problem! We agree with the reviewer that the theoretical analyses around the GSE problem (more broadly, connections among various safe RL formulations) are an interesting research direction. We receive the reviewers' comments seriously and will consider them in future work.
> > >
> > > **Comparison with [Ref-3], [Ref-4], and [28].** We agree with the reviewer that our MASE algorithm for probability one constraint can be regarded as a nice generalization of [28] combined with a nice variant of shielding methods, which we also believe is an interesting and useful connection in the safe RL community. As we responded to Reviewer sDLd, we will surely add such discussion in the camera-ready paper.
> > >
> > > **Boxplots.** Thank you for seeing the new experimental results in a new one-page PDF. We agree with the reviewer that the boxplots are important for a fair comparison to [28]; hence, we will add them to the final version of the main paper (not to the appendix).
> > >
> > > We appreciate your valuable suggestions and feedback for improving the quality of our manuscript.

---

### Official Review · Reviewer_Q24U · 2023-07-05

**Soundness:** 3 good
**Presentation:** 3 good
**Contribution:** 3 good
**Rating:** 5
**Confidence:** 3

**Summary:**

Paper considers the generalized safe exploration problem (problem 4) and compares it to the other safe exploration problems in the literature, leading to Thm 3.1 that concludes that it is more general than the others. The authors then introduce MASE, a meta-algorithm for safe exploration, that attempts to solve the GSE problem with the ability to execute an emergency stop "beforehand" as opposed to others that have done it afterwards. Section 6 then presents a more practical algorithm using GP models, which is then used in section 7 to compare the new approach with several baselines.

**Strengths:**

The paper is well presented, with several technical results about the safety analysis, culminating in Thms 5.6 and 5.7 which show that GLM-MASE guarantees safety with high probability for every time step.

Several numerical experiments are presented to compare with unconstrained and SOA constrained RL algorithms.

**Weaknesses:**

The development in the paper assumes access to an emergency stop action that enables the agent to avoid violating a safety constraint. This assumptions seems difficult to achieve in practice as for many agents of interest, stopping is not a safe state. Further, the action would typically be state dependent, requiring at a minimum a emergency action policy (rather than action). Finding either action or policy seems non trivial and could possibly be overly simplifying most problems of interest.

Given the comment about [33] at the bottom of page 4, I would have expected to see a comparison of these approaches to imposing an action to avoid an unsafe state before/after a training epoch in the numerical results. If that is included in section 7, suggest highlight that discussion point more.

The footnote on page 3 makes reference to this being a conservative approximation of that in safe RL problems with chance constraints, and consigns the discussion Appendix B. It would seem like more clarification than that is needed here. Also, the more recent work by Pavone in this area will be of interest:
* Lucas Janson, Edward Schmerling, and Marco Pavone, “Monte Carlo motion planning for robot trajectory optimization under uncertainty.” In Robotics Research, pages 343–361. Springer, 2018.
* Anirudha Majumdar and Marco Pavone “How Should a Robot Assess Risk? Towards an Axiomatic Theory of Risk in Robotics,” https://doi.org/10.48550/arXiv.1710.11040

Fig 2 shows that MASE satisfies the constraints, but if I understand the plot correctly, this is achieved with very conservative margins (constraint at 20, values are typical ≤ 5). This is similar to Saute TRPO, but perhaps suggest why the performance (episode return) is so weak compared to the unconstrained solutions (that don’t violate the constraints by much). Is there a way to trade off  this conservatism to achieve better performance?

Very hard to see the frequency with which TRPO Lagrangian violates the constraints given the lines/colors on fig 2.

**Questions:**

Does the comment about safe RL problems with chance constraints suggest that there is a better answer available already? The text says that problem 5 in App B is hard to solve, but presumably the authors of [20] and [22] did so? Can their results be compared to the ones here?

**Limitations:**

See little discussion of this point in the paper.

---

> ### Author Rebuttal · Authors · 2023-08-08
>
> We deeply appreciate the reviewer’s valuable comments and feedback.
>
> We will answer the Questions first and then address the comments provided in Weakness.
>
> **Chance constraint [Question and Weakness 3].** Thank you for the good question! In general, it is known that joint chance constraints represented in Problem 5 are hard to handle. Most of the previous work does not directly deal with this type of constraint and uses some approximations or assumptions. For example, Pfrommer et al. [22] assume a known linear time-invariant dynamics. Also, One et al. [20] approximates the joint chance constraint as follows:
>
>
> \begin{align}
>         \Pr \left[ \bigvee_{h=1}^{H} s_h \in S_\text{unsafe} \mid P, \pi \right] \le \mathbb{E} \left[ \sum_{h=1}^H  \mathbb{I}(s_h \in S_\text{unsafe}) \mid P, \pi \right].
> \end{align}
>
> This is a conservative approximation with an additive structure, which is easier to solve than the original joint chance constraint. One et al. [20] deals with the above constraints with additive safety structure. Remark B.3 implies that, by additionally transforming the conservatively-approximated problem into the GSE problem, the problem would become easier to handle because the safety constraint is instantaneous. As for the comparison with [20] and [22], they assume (partially) known system dynamics; thus, it is not possible to directly compare our method and theirs. We appreciate the reviewer’s comments and questions and the good pointers. We will add more discussion mentioned above in the next version based on the reviewer’s comments.
>
> **Emergency stop action [Weakness 1].** As discussed in Appendix I, emergency stop actions should be avoided in some cases. However, the biggest objective of this paper is to 1) formulate the GSE problem and 2) propose the MASE algorithm for solving it. Emergency stop action is a variant of resetting actions, which have been commonly used in episodic RL literature. The issue in the resetting action has been discussed in general RL problem settings, which have been addressed by many existing RL studies as being represented by [13]. We consider that it is not very difficult to combine our proposed method with such previous work (although we additionally need to incorporate the safety budget to return to the initial state). To clearly convey the core ideas or contributions of our approach, we consider that it is necessary and reasonable to introduce the emergency stop action.
>
> **Early termination [Weakness 2].** Thank you for the good comments. **We conducted an experiment and presented new results in a PDF provided in the global response.** The early-terminated MDP (ET-MDP, [33]) and our MASE exhibit similar learning curves on the average episode reward and average episode safety. However, while ET-MDP violated the safety constraint in most episodes (i.e., almost all episodes are terminated after an unsafe action is executed), our MASE did *not* violate any safety constraint.
>
> **Performance [Weakness 4].** We appreciate valuable comments. As the reviewer mentions, our proposed algorithm is sometimes very conservative since it seeks to guarantee the safety constraint at every time step and episode, while leveraging the uncertainty quantifier. This is the reason why our algorithm performs worse than CPO or TRPO-Lagrangian that encourages safety in a more loose manner, in terms of reward. However, we consider that the main objective of the experiment is to show the validity of our GSE problem and MASE algorithm for which we need to prove that the safety constraint is satisfied empirically consistently with the theory. As the reviewer points out, it would be an important and interesting direction to consider how to balance the trade-off between safety and reward in our GSE problem and MASE algorithm. Although we provide a theoretical result on optimality (Theorem 5.5) under the generalized linear CMDP assumption, we have observed that it is difficult to achieve comparable reward performance to safety-agnostic RL algorithms (e.g., TRPO) or safe RL algorithms with loose constraints (e.g., CPO, TRPO-Lagrangian) in complicated tasks. We would like to leave this issue in the future work. We deeply appreciate the reviewer’s valuable comments.
>
> **TRPO in Figure 2 [Weakness 5].** Thank you for the advice to improve the presentation. We will modify the color and line width so that it is easier to see the result of TRPO-Lagrangian.
>
> We thank the reviewer for the time and effort on reviewing our paper.
>
> ---
> References
>
> [13] Eysenbach, B, et al. "Leave no trace: Learning to reset for safe and autonomous reinforcement learning." ICLR (2017).
>
> [20] Ono, M., Pavone, M., Kuwata, Y., and Balaram, J. (2015). Chance-constrained dynamic programming with application to risk-aware robotic space exploration. Autonomous Robots,39(4):555–571.
>
> [22] Pfrommer, S., Gautam, T., Zhou, A., and Sojoudi, S. (2022). Safe reinforcement learning with chance-constrained model predictive control. In Learning for Dynamics and Control Conference (L4DC), pages 291–303.
>
> [33] Sun, H., Xu, Z., Fang, M., Peng, Z., Guo, J., Dai, B., and Zhou, B. (2021). Safe exploration by 455 solving early terminated MDP. arXiv preprint arXiv:2107.04200.

---

> > ### Comment · Reviewer_Q24U · 2023-08-14
> > **Reply**
> >
> > Authors have mostly addressed my concerns and I will raise my score accordingly

---

> > > ### Author Response · Authors · 2023-08-15
> > > **Responses to further comments.**
> > >
> > > We appreciate your valuable suggestions and thank you for increasing the score! We will ensure that all reviewers' valuable feedback is reflected in the camera-ready paper.

---

### Official Review · Reviewer_sDLd · 2023-07-10

**Soundness:** 3 good
**Presentation:** 2 fair
**Contribution:** 1 poor
**Rating:** 5
**Confidence:** 4

**Summary:**

This paper addresses the problem of safe reinforcement learning, in particular safe exploration. In a nutshell, the authors provide an algorithm that is supposed to go beyond standard 'safety' measures in safe RL, such as the constrained MDP setting. There, an agent is bound to satisfy an additional (expected) cost constraint. Here, the authors postulate that an agent must satisfy a constraint almost surely or with high probability. As a key feature of their approach, the authors assume that a so-called  'emergency stop action' is available that allows the agent to always have a fallback. The authors provide a general framework in the form of an algorithm, prove its theoretical guarantees, and evaluate the method on a set of standard benchmarks.

**Strengths:**

The authors tackle a very important problem, safe exploration in RL. Moreover, they identify correctly that the standard constrained RL (or constrained MDP) setting is generally insufficient to ensure safety during exploration. In principle, the constrained RL setting provides just an incentive to act safely during training, and even after training, safety depends on an expectation to satisfy a safety constraint. Generally, the paper is well-written and easy to follow.

**Weaknesses:**

I like the paper in general, but in its current state cannot be accepted, in my opinion. The reason is a severe lack of related work. Most importantly, the authors seem unaware of a flavor of safe RL that is often referred to as 'shielding.' In these settings, a so-called shield 'blocks' unsafe actions according to some pre-defined safety measure. [1] was the first paper to introduce this aspect, [2] introduced shields for almost-sure properties in partially observable environments, [3] provides shields that satisfy a property with a certain probability, [4] provides a shielding mechanism for multi-agent settings. There are many more relevant works. The general shielding framework depends on varying assumptions, but the general procedure looks like the MASE algorithm. Note that safety during exploration/training is the most important motivation for shielding. I encourage the authors to thoroughly compare these (and more) works to explain the contribution and novelty better.

[1] Alshiekh et al.: Safe Reinforcement Learning via Shielding. AAAI 2018

[2] Carr et al.: Safe Reinforcement Learning via Shielding for POMDPs. AAAI 2023

[3] Jansen et al.: Safe Reinforcement Learning Using Probabilistic Shields. CONCUR 2020

[4] Melcer et al.: Shield Decentralization for Safe Multi-Agent Reinforcement Learning. NeurIPS 2022

Moreover, I am not convinced by the experimental evaluation. The emergency action seems central to the approach, but I fail to see how it has been integrated into the standard benchmarks. Then, how often is it called during training by the agent? How does it impede an agent's exploration rate?


minor comments

l92: In the definition of a policy, it seems to be non-stochastic. In such a multi-objective setting, it might be beneficial to use stochastic policies. Have you considered this?

l 125/126: in a probabilistic setting, there might be states with a high probability of violating a safety constraint but do not already violate it. It could be interesting to consider such information in the value function.

**Questions:**

1. Compare your approach to the shielding framework.

2. How is the emergency action affecting the agent, please discuss in line with the experiments.

3. How is the 'blocking' of actions realized for an RL agent on a technical level?

**Limitations:**

The limitations are not properly addressed, see my earlier comments on the evaluation.

---

> ### Author Rebuttal · Authors · 2023-08-08
>
> We deeply appreciate the reviewer for helpful and thoughtful comments and questions.
>
> First of all, please let us emphasize that **our main contributions include the formulation of the GSE problem and its theoretical result (i.e., Theorem 3.1)** as well as the proposal of the MASE algorithm. Our contribution regarding the MASE algorithm is not only on the soundness or novelty of the algorithm itself but also the applicability to a wide range of safe RL problems such as Problems 1, 2, and 3, which is supported by the good theoretical property of the GSE problem backed by Theorem 3.1.
>
> We will first answer the Questions and then address the comments in Weakness. We will provide responses together for the similar Questions and comments written in Weakness.
>
> **Shielding method [Question 1 and Weakness 1].** Thank you for pointing out related work. We have read all the papers the reviewer raises as examples and notable ones on shielding. As the reviewer mentions, the notion of emergency stop actions is akin to the shielding. We now consider that it would be reasonable to regard our MASE algorithm as a variant of shielding methods (especially, preemptive shielding in [Ref-1]) that is specialized for the GSE problem. On the other hand, the MASE algorithm does not only block unsafe actions but also provides proper penalty for executing the emergency stop actions based on the uncertainty quantifier. Thus, this algorithm provides rigorous theoretical guarantees on optimality (e.g., Theorems 4.2 and 5.7) as well as safety (e.g., Theorems 4.1, 5.6, and 6.1). In particular, under the generalized linear CMDP assumption, our proposed MASE algorithm provides theoretical guarantees on both safety and optimality, which is a unique property from the perspective of safe RL research based on shielding. We can enjoy this theoretical advantage in many safe RL problems because of the wide applicability of the GSE problem. We will discuss the relationship between the shielding method and our MASE algorithm, while properly citing the papers mentioned by the reviewer (i.e., [Ref-2], [Ref-3], [Ref-4]).
>
> In addition, thanks to the reviewer’s comments, we found that our GSE problem may contribute to bridging the gap between shielding methods and other safe RL methods. We appreciate the constructive feedback.
>
> **Experiment [Question 2-3 and Weakness 2]**. In our experiment, when the agent identified that there was no safe action based on the GP-based uncertainty quantifier, we simply terminated the current episode (i.e., resetting) immediately after the emergency stop action and started the new episode. Also, the frequency of the emergency stop actions is:
>
> |  Task |  total  |  Last 100 epochs  |
> | ---- | ---- | ---- |
> |  PointGoal1  |  154/500  |  24/100  |
> |  CarGoal1  |  397/500  |  46/100  |
>
> The emergency stop is a variant of so-called resetting actions that are quite common in episodic RL settings, which actually prevent the agent from exploring the state-action spaces since the uncertainty quantifier is sometimes quite conservative. We consider that this is the reason why the reward performance of our MASE is worse than other methods (e.g., TRPO-Lagrangian, CPO) in Figures 2a and 2d. However, because we require the agent to solve more difficult problems where safety is guaranteed at every time step and episode, we consider that this result is reasonable to some extent. Though it is better for an algorithm for such a severe safety constraint to have a comparable performance as CPO, we will leave it for future work. The aforementioned discussion will be useful for the readers, so we will add it to the experiment section in the camera-ready version.
>
>
> **Stochastic policy [Minor comments 1]**. To clarify the contributions of our paper, we focus on the deterministic policy. As the reviewer mentions, however, stochastic policy settings would be beneficial in many cases. We consider that our two core ideas (the GSE problem and the MASE algorithm) are quite simple and intuitive, which can be extended to stochastic policy settings, but the mathematics would be much more complicated.
>
> **Value function incorporating safety [Minor comments 2]**. Thank you for the great advice! It seems a promising idea to pessimistically estimate the value function for the states that are likely to violate the safety constraint.
>
> We will improve our paper in the final version based on the reviewer’s comments especially on the connections with the shielding method.
>
> ---
> References
>
> [Ref-1] Alshiekh et al.: Safe Reinforcement Learning via Shielding. AAAI 2018
>
> [Ref-2] Carr et al.: Safe Reinforcement Learning via Shielding for POMDPs. AAAI 2023
>
> [Ref- 3] Jansen et al.: Safe Reinforcement Learning Using Probabilistic Shields. CONCUR 2020
>
> [Ref-4] Melcer et al.: Shield Decentralization for Safe Multi-Agent Reinforcement Learning. NeurIPS 2022

---

> > ### Comment · Reviewer_sDLd · 2023-08-11
> > **Thanks**
> >
> > I thank the authors for their careful reply. With the new experimental results and the proper placement in the literature, I will increase my score.

---

> > > ### Author Response · Authors · 2023-08-11
> > > **Thank you for additional reply**
> > >
> > > We would like to express our sincere gratitude to the reviewer who read through our responses and will raise the score. We believe that the valuable comments from the Reviewer sDLd are very helpful for us to improve our paper. Thank you very much for your considerable comments.

---

> > > > ### Author Response · Authors · 2023-08-14
> > > >
> > > > Though authors are not allowed to submit a revised version per NeurIPS-23 policy until the camera-ready submission, we have already revised the paper based on the reviewers' comments. The major concerns by Reviewer sDLd have been addressed in the revised version as follows:
> > > >
> > > > - We added the following paragraph between lines 209 and 210 to discuss the relationships between shielding and MASE.
> > > >
> > > >    > **Connections to shielding methods.** The notion of the emergency stop action is akin to shielding [2 , 20] which has been active studied in various problem settings including partially-observable environments [10] or multi-agent settings [23]. Thus, MASE algorithm can be regarded as a variant of shielding methods (especially, preemptive shielding in [2]) that is specialized for the GSE problem. On the other hand, the MASE algorithm does not only block unsafe actions but also provides proper penalty for executing the emergency stop actions based on the uncertainty quantifier, which leads to rigorous theoretical guarantees presented shortly. Such theoretical advantages can be enjoyed in many safe RL problems because of the wide applicability of the GSE problem backed by Theorem 3.1.
> > > >
> > > > - References are added as follows (numbers changed by adding new references):
> > > >     - [2] Alshiekh, M., Bloem, R., Ehlers, R., Könighofer, B., Niekum, S., and Topcu, U. (2018). Safe reinforcement learning via shielding. In AAAI Conference on Artificial Intelligence (AAAI).
> > > >     - [10] Carr, S., Jansen, N., Junges, S., and Topcu, U. (2023). Safe reinforcement learning via shielding under partial observability. In AAAI Conference on Artificial Intelligence.
> > > >     - [20] Könighofer, B., Bloem, R., Junges, S., Jansen, N., and Serban, A. (2020). Safe reinforcement learning using probabilistic shields. In International Conference on Concurrency Theory: CONCUR.
> > > >     - [23] Melcer, D., Amato, C., and Tripakis, S. (2022). Shield decentralization for safe multi-agent reinforcement learning. In Neural Information Processing Systems.
> > > >
> > > > - We have also added the new experimental results including the new figures (in the new one-page PDF) and the new experimental results (presented to Reviewer sDLd) to reflect the reviewers' feedback.
> > > >
> > > > We deeply appreciate the helpful comments from the Reviewer sDLd for improving the quality of our manuscript.

---

> > > > > ### Author Response · Authors · 2023-08-20
> > > > >
> > > > > As the discussion period draws to a close, we would like to kindly request to consider increasing the score based on the following initial reply provided by the reviewer, as other reviewers have conducted.
> > > > >
> > > > > > With the new experimental results and the proper placement in the literature, I will increase my score.
> > > > >
> > > > > Should there be any remaining concerns, we remain open to addressing any concerns regarding our work.
> > > > >
> > > > > We sincerely appreciate your time and effort in reviewing our work.

---

### Author Rebuttal · Authors · 2023-08-08

Dear reviewers and AC,

We deeply thank all the reviewers for their insightful comments and constructive suggestions.

- We have conducted new experiments based on the reviewers' comments. Additional experimental results are provided in a one-page PDF containing new figures attached in this "global" response.

- We have provided our detailed response to each reviewer with a separate response.

We hope our replies have addressed all the questions and concerns of the reviewers. We are willing to answer any of the reviewers' concerns about our work and sincerely wish the reviewers to value the technical innovation and overall contributions of our paper.

Best regards,

Authors.

---

### Decision · Program_Chairs · 2023-09-21

**Decision:**

Accept (poster)

**Comment:**

The paper defines a novel "generalized safe exploration" problem and gives a meta-algorithm for the problem that combines an unconstrained RL algorithm with an uncertainty quantifier. The problem is interesting and the technical results are sound and novel. While there were some concerns about the discussion of related work and experiments in the original paper, the new results and clarifications added during the discussion period addressed them. Given all this, I am recommending acceptance.  Please incorporate the reviewers' feedback carefully in the final version.